# Acetylation of KLF5 maintains EMT and tumorigenicity to cause chemoresistant bone metastasis in prostate cancer

Baotong Zhang [1,2], Yixiang Li[1,2], Qiao Wu[3], Lin Xie[1,2,4], Benjamin Barwick [1,2], Changying Fu[3,5], Xin Li[6], Daqing Wu[6], Siyuan Xia[1,2], Jing Chen[1,2], Wei Ping Qian[2,7], Lily Yang[2,7], Adeboye O. Osunkoya[2,8], Lawrence Boise [1,2], Paula M. Vertino [2,9], Yichao Zhao[1,2], Menglin Li[1,2], Hsiao-Rong Chen[10], Jeanne Kowalski[10], Omer Kucuk[1,2], Wei Zhou [1,2] & Jin-Tang Dong [1,2,5✉]

Advanced prostate cancer (PCa) often develops bone metastasis, for which therapies are very limited and the underlying mechanisms are poorly understood. We report that bone-borne TGF-β induces the acetylation of transcription factor KLF5 in PCa bone metastases, and acetylated KLF5 (Ac-KLF5) causes osteoclastogenesis and bone metastatic lesions by activating CXCR4, which leads to IL-11 secretion, and stimulating SHH/IL-6 paracrine signaling. While essential for maintaining the mesenchymal phenotype and tumorigenicity, Ac-KLF5 also causes resistance to docetaxel in tumors and bone metastases, which is overcome by targeting CXCR4 with FDA-approved plerixafor. Establishing a mechanism for bone metastasis and chemoresistance in PCa, these findings provide a rationale for treating chemoresistant bone metastasis of PCa with inhibitors of Ac-KLF5/CXCR4 signaling.

[1] Department of Hematology and Medical Oncology, Emory University School of Medicine, Atlanta, GA, USA. [2] Winship Cancer Institute, Emory University, Atlanta, GA, USA. [3] Department of Genetics and Cell Biology, Nankai University College of Life Sciences, Tianjin, China. [4] Third Affiliated Hospital of Kunming Medical University, Cancer Hospital of Yunnan Province, Kunming, China. [5] Department of Human Cell Biology and Genetics, Southern University of Science and Technology School of Medicine, Shenzhen, China. [6] Molecular Oncology and Biomarkers Program, Georgia Cancer Center, Department of Biochemistry and Molecular Biology, Medical College of Georgia, Augusta University, Augusta, GA, USA. [7] Department of Surgery, Emory University School of Medicine, Atlanta, GA, USA. [8] Department of Pathology and Urology, Emory University School of Medicine, Atlanta, GA, USA. [9] Department of Radiation Oncology, Emory University School of Medicine, Atlanta, GA, USA. [10] Department of Biostatistics & Bioinformatics, Rollins School of Public Health, Emory University, Atlanta, GA, USA. ✉email: dongjt@sustech.edu.cn

One in five men will be diagnosed with prostate cancer (PCa) in his lifetime[1]; and ~10–20% of prostate cancers progress to castration-resistant prostate cancer (CRPC)[2]. Approximately 90% patients with metastatic CRPC (mCRPC) develop bone metastases[3,4]. Although denosumab or zoledronic acid is recommended for CRPC patients with bone metastases to prevent or delay skeletal-related events (SREs)[5,6], no significant benefits in terms of cancer-specific and overall survival have been observed[7]. At present, the taxanes docetaxel and cabazitaxel are the only chemotherapeutic agents that have a survival benefit for mCRPC patients, but virtually all mCRPCs eventually develop resistance[8], and docetaxel-treated patients with bone metastasis still have poor prognoses[9]. Effective therapeutic strategies are still lacking for patients with chemoresistant bone metastases.

Although PCa bone metastases are often characterized as osteoblastic on CT images, both osteoclastic and osteoblastic activities are involved in the formation of bone metastatic lesions[10–12]. The development and outgrowth of bone metastases from PCa cells is a multi-step process in which tumor cells and other cells in the bone microenvironment communicate with each other via paracrine factors and receptors on the cell surfaces[13]. For instance, paracrine signaling from tumor cells such as PTHrP stimulates osteoblasts to produce RANKL[14], which promotes the differentiation of osteoclasts (i.e., osteoclastogenesis) and initiates bone resorption[15]. Osteolytic resorption further releases active factors, including TGF-β, that support cancer cell survival and growth, which in turn secrete more osteolytic factors such as PTHrP, IL-11, VEGF, and MMPs[15–19]. This vicious cycle between TGF-β and osteolytic resorption is one driving force for PCa bone metastasis, and targeting TGF-β signaling may thus lead to therapies against bone metastasis[20,21]. On the other hand, protein expression analysis reveals the occurrence of EMT in bone metastasis[22], and cancer cells in the bone marrow are often resistant to chemotherapy because they are held in the G0 phase of the cell cycle by contacting with bone marrow stromal cells[15]. Although recent studies have begun to decipher the molecular mechanisms underlying bone metastatic growth of PCa, the process is still poorly understood and effective therapies remain to be developed. Crucial mediators that contribute to bone metastasis and chemoresistance are particularly interesting because some can be therapeutic targets.

TGF-β is usually produced by tumor microenvironments, but advanced tumors also acquire an autocrine ability to secrete TGF-β[23]. Although TGF-β suppresses tumor growth during the early stages of tumorigenesis, it is a potent inducer of epithelial–mesenchymal transition (EMT) and bone metastasis in advanced tumors[24–29]. TGF-β also induces chemoresistance in multiple types of cancers, including PCa[30–34]. In addition, TGF-β-induced EMT also generates stem cell properties[35], which is likely involved in tumor progression. The dual roles of TGF-β in tumor development and progression have been one of the major challenges in the development of therapeutic agents targeting the TGF-β pathway[36,37].

Our previous studies have demonstrated that TGF-β induces the acetylation of transcription factor Krüppel-like factor 5 (KLF5) at lysine 369 (K369) via SMAD-recruited p300 acetylase[38,39], and acetylated KLF5 (Ac-KLF5) then forms a transcriptional complex different from that of deacetylated KLF5 (deAc-KLF5), which reverses the functions of deAc-KLF5 and is essential for TGF-β to function in gene regulation, cell proliferation, differentiation, and tumorigenesis[38–42]. Considering the established TGF-β/Ac-KLF5 signaling axis and the function of TGF-β in the induction of EMT, bone metastases, and chemoresistance, we hypothesize that Ac-KLF5 plays a crucial role in the development of chemoresistant bone metastases in PCa.

In this study, we tested this hypothesis in multiple in vitro and in vivo models, and found that Ac-KLF5 was highly expressed in bone metastases of human PCa and mouse models. Tumor cells expressing the Ac-KLF5-mimicking mutant KLF5$^{K369Q}$ (KLF5$^{KQ}$ or KQ) caused bone metastatic lesions and became resistant to docetaxel in mouse models while maintaining a mesenchymal phenotype and tumorigenicity. Mechanistically, KLF5$^{KQ}$ transcriptionally activated *CXCR4* expression, which in turn increased the secretion of IL-11 to promote osteoclast differentiation. Importantly, combined treatment with docetaxel and the CXCR4 inhibitor plerixafor effectively suppressed KLF5$^{KQ}$-induced bone metastatic lesions and sensitized KLF5$^{KQ}$-expressing tumors to docetaxel. These findings identify an acetylation-modified transcription factor KLF5 as a key modulator of tumor cell plasticity in the development and outgrowth of PCa bone metastases, and provide a rationale for using docetaxel plus plerixafor in the treatment of PCa bone metastases expressing Ac-KLF5.

## Results

**Acetylation of KLF5 induced by TGF-β occurs more extensively in osteolytic than subcutaneous tumors.** Osteoclastic bone resorption during the bone metastasis of PCa releases TGF-β, a paracrine factor that drives a vicious cycle of cancer growth in bone[43]. Blockage of TGF-β signaling by SD-208, an inhibitor of TGF-β receptor I (TGFBR1), effectively reduces PCa bone metastasis[44]. We have shown that TGF-β induces the acetylation of KLF5 at K369, and acetylated KLF5 (Ac-KLF5) is essential for TGF-β to regulate gene expression and cell proliferation[38–41], which has established Ac-KLF5 as a functional effector of TGF-β. We thus investigated whether and how Ac-KLF5 plays a role in TGF-β-induced bone metastasis of PCa.

First, we tested whether prostate tumors grown in the bone express more Ac-KLF5 than those grown subcutaneously. Interestingly, this was indeed the case, as IHC staining of Ac-KLF5 demonstrated that tibial tumors from both DU 145 and PC-3 PCa cell lines indeed expressed significantly higher levels of Ac-KLF5 than their counterparts, while total KLF5 expression levels were lower in tibial tumors (Fig. 1a–f).

When TGF-β signaling was blocked by SD-208, an effective inhibitor of TGF-β receptor I, subcutaneous tumor growth was promoted (DU 145 cells, Supplementary Fig. 1a–c) or unaffected (PC-3 cells, Supplementary Fig. 1d–f), while tibial tumor growth was suppressed in both DU 145 and PC-3 cell lines (Supplementary Fig. 1g–i). The TGFBRI inhibitor SD-208 dramatically reduced the expression of Ac-KLF5 in both tibial and subcutaneous tumors of both cell lines (Fig. 1a, b, c, e) and restored total KLF5 expression in tibial tumors (Fig. 1a, b, d, f). Consistently, in cultured PC-3, DU 145 and C4-2B cells, TGF-β treatment also increased Ac-KLF5 expression level while decreasing total KLF5 expression (Fig. 1g–i and Supplementary Fig. 1j), rendering an increase in the ratio of Ac-KLF5 to total KLF5. Therefore, TGF-β induced significant acetylation of KLF5 in PCa cells during their metastatic growth in the bone.

**Acetylation of KLF5 is required for TGF-β to induce EMT.** TGF-β is a master regulator of EMT[45], a process believed to promote tumor metastasis[25]. We thus tested whether acetylation of KLF5 is involved in EMT. Accompanying the induction of KLF5 acetylation, EMT features were also enhanced in tibial tumors, as indicated by reduced E-cadherin expression and increased vimentin expression (Supplementary Fig. 1k). The EMT phenotype in tibial tumors was also coupled with TGF-β activity, as the SD-208 inhibitor of TGF-β signaling apparently upregulated epithelial marker E-cadherin while downregulating mesenchymal marker vimentin (Supplementary Fig. 1k).

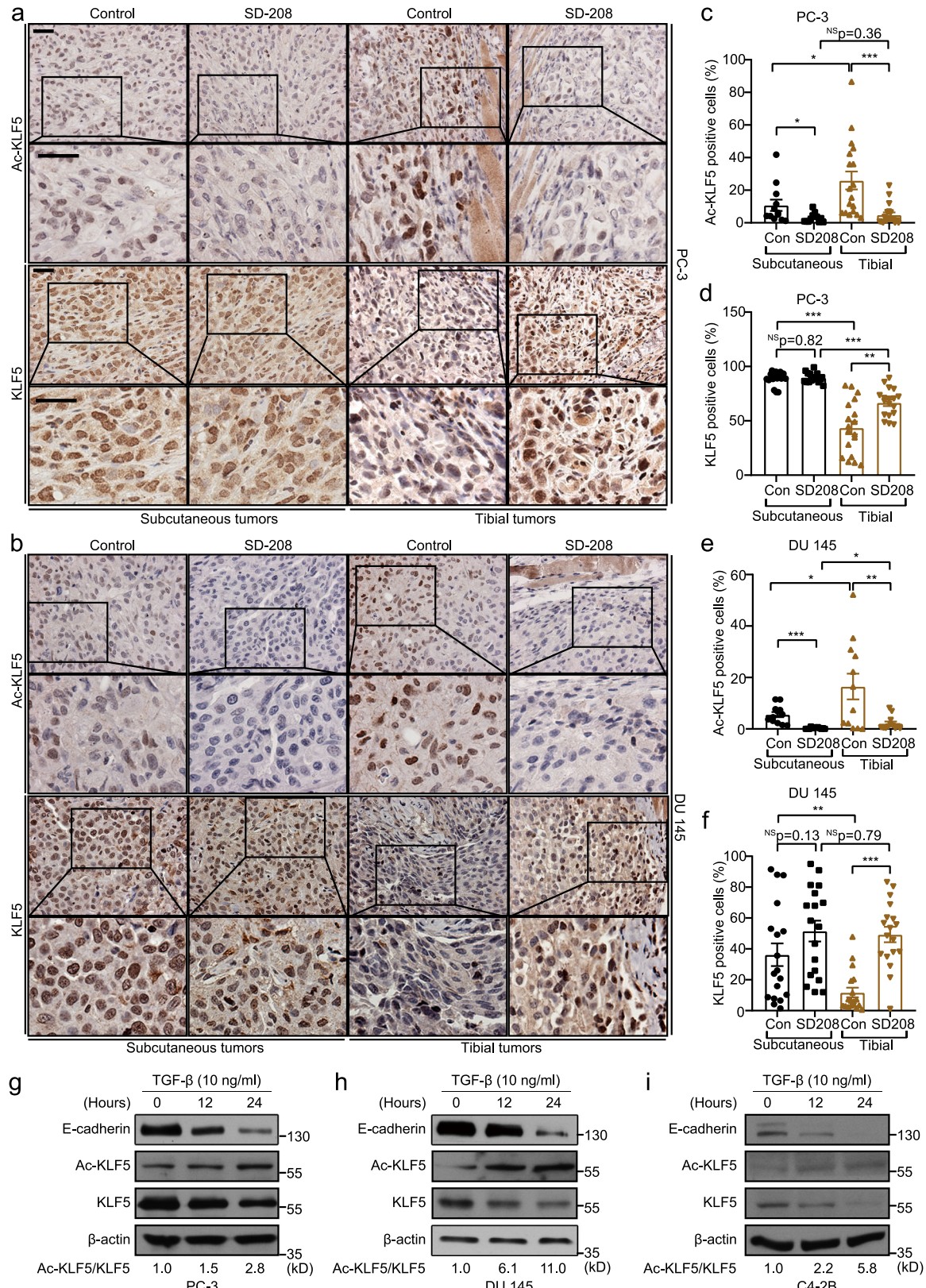

We also confirmed that TGF-β treatment induced EMT features in vitro, including spindle-like morphology (Supplementary Fig. 1l), decreased E-cadherin, and increased mesenchymal markers N-cadherin, FN1, and ZEB1 (Supplementary Fig. 1m).

To determine whether the induction of EMT by TGF-β depends on the acetylation of KLF5, we ectopically expressed wild-type KLF5, acetylation-deficient mutant KLF5K369R (KLF5KR or KR), and acetylation-mimicking mutant KLF5K369Q (KLF5KQ or KQ) in DU 145 parental cells, and then examined their effects on the induction of EMT by TGF-β. Interestingly, the KLF5KQ mutant itself induced EMT, as indicated by lower E-cadherin, higher vimentin and enhanced migration and invasion

**Fig. 1 TGF-β enriched in the bone induces acetylation of KLF5 at K369. a, b** IHC staining of Ac-KLF5 and KLF5 in subcutaneous and tibial tumors of PC-3 cells (**a**) and DU 145 cells (**b**) from mice treated with the SD-208 TGF-β inhibitor (50 mg/kg/day). Scale bars, 50 µm. **c–f** Statistical analysis of the IHC images in **a** and **b** by calculating the percentages of Ac-KLF5 (**c, e**) or KLF5 (**d, f**) positive cells in subcutaneous and tibial tumors of PC-3 (**c, d**) or DU 145 (**e, f**) cells, as counted by Fiji software. For each condition in **c–f**, three tumors from three mice were used, and $n = 18$ different images from the three tumors were analyzed, except for Ac-KLF5 in subcutaneous PC-3 and DU 145 tumors and tibial DU 145 tumors, where $n = 12$ different images were used. Scatter bars in black indicate subcutaneous tumors and those in brown indicate tibial tumors. **g–i** Detection of indicated proteins by western blotting in whole cell lysates of TGF-β treated PC-3 (**g**), DU 145 (**h**), and C4-2B (**i**) cells in in vitro two-dimentional culture. The ratio of Ac-KLF5 to total KLF5 is indicated below the blots. Western blotting assays were repeated at least twice and consistent results were achieved as shown in Supplementary Fig. 1j. In panels **c–f**, data are shown in mean ± S.E.M. *$p < 0.05$; **$p < 0.01$; ***$p < 0.001$; NS not significant (two-tailed Student's $t$ test). Source data are provided as a Source Data file.

capabilities (Supplementary Fig. 2a–d). Furthermore, the KLF5$^{KQ}$ mutant maintained but the KLF5$^{KR}$ mutant abolished TGF-β-induced EMT, as indicated by the expression of EMT markers E-cadherin and vimentin, morphological changes, and the migration and invasion capability (Supplementary Fig. 2a–d). These findings indicate that acetylation of KLF5 is essential for TGF-β to induce EMT in PCa cells.

**Acetylation of KLF5 causes bone metastatic lesions while maintaining the mesenchymal phenotype in tibial tumors.** We further investigated whether acetylation of KLF5 is required for bone metastatic growth of PCa. We deleted the endogenous *KLF5* to generate KLF5-null (KLF5$^{-/-}$) PC-3, DU 145, and C4-2B cells by introducing deletions that lead to early termination using the CRISPR-Cas9 system (Supplementary Fig. 3a–f), and then stably expressed empty vector (EV), wild-type *KLF5* (*KLF5*), *KLF5$^{KR}$* (*KR*), and *KLF5$^{KQ}$* (*KQ*; Supplementary Fig. 3g) to establish PCa cells with different KLF5 acetylation statuses. These cells were injected into the tibias of nude mice. Interestingly, loss of KLF5 or interruption of KLF5 acetylation at K369 significantly attenuated tumor formation in the tibia, as the incidence of tumor formation for KLF5$^{-/-}$ or KLF5$^{KR}$ was less than that for KLF5 or KLF5$^{KQ}$ (Fig. 2a). Consistently, tumor cells expressing KLF5 or KLF5$^{KQ}$ gave rise to more severe bone lesions than those expressing KLF5$^{KR}$ or EV as indicated by radiographs of tumor-bearing tibias (Fig. 2b). Moreover, cells with KLF5 or KLF5$^{KQ}$ induced larger tumor areas in the bone than those with KLF5$^{-/-}$ or KLF5$^{KR}$, as indicated by H&E staining of bones with tumors and quantitative analyses of tumor areas (Fig. 2c, d). Notably, PCa cells lacking KLF5 (EV) hardly formed tumors in the bone (Fig. 2c, d). On the other hand, cells carrying KLF5 and KLF5$^{KQ}$ caused comparable tibial tumors, consistent with KLF5 acetylation occurring more extensively in tibial tumors due to enriched TGF-β in the bone (Fig. 1c, e). These findings demonstrate that acetylation of KLF5 at K369 plays an indispensable role in bone metastatic growth of PCa cells.

Considering the role of Ac-KLF5 in EMT, we also detected the EMT markers E-cadherin and vimentin in tibial tumors. KLF5$^{KR}$ tumors in the tibia still maintained their epithelial characteristics, including stronger E-cadherin expression and weaker vimentin expression (Fig. 2e, f). In contrast, KLF5$^{KQ}$ tumors exhibited weaker E-cadherin at cell junctions and a much stronger staining for mesenchymal marker vimentin (Fig. 2e, f). Therefore, Ac-KLF5 also appears to be required for tumor cells in the bone to maintain a mesenchymal phenotype.

Although KLF5-null (KLF5$^{-/-}$) cells had negligible (PC-3 and DU 145) or significantly slowed (C4-2B) tumor growth in the bone (Fig. 2a–d), they were able to propagate in vitro. Consistent with the induction of EMT by the knockdown of *KLF5* in immortalized epithelial cells[46], deletion of *KLF5* in PC-3 PCa cells induced EMT even without TGF-β, as indicated by the mesenchymal morphology, downregulation of epithelial marker E-cadherin, upregulation of mesenchymal markers N-cadherin

and vimentin, and enhanced cell migration and invasion (Supplementary Fig. 3h–j). Furthermore, expression of wild-type KLF5 and KLF5$^{KR}$ restored the epithelial phenotypes but expression of KLF5$^{KQ}$ maintained mesenchymal features, including spindle-like morphology, lower E-cadherin expression, higher vimentin expression, and enhanced migration and invasion capability (Supplementary Figs. 3k–m and 4a–d). Therefore, acetylation of KLF5 is also required for mesenchymal features in vitro. TGF-β was added to the medium for cultured DU 145 cells to mimic a TGF-β-enriched environment like bone, and we found that TGF-β decreased E-cadherin at the cell junctions and enhanced vimentin and cell motility in DU 145 cells with wildtype KLF5 but not in those with KLF5$^{KR}$ mutant, rendering an EMT phenotype in cells with wild-type KLF5, which was similar to cells with the KLF5$^{KQ}$ mutant (Supplementary Fig. 4a–d) and further suggests that TGF-β-induced KLF5 acetylation at K369 is necessary for EMT induction.

Expression of the proliferation marker Ki67 was low in KLF5-null tibial tumors (Fig. 2g, h), consistent with the necessity of KLF5 for tumor growth in the tibia. Unexpectedly, KLF5$^{KR}$ tibial tumors had a significantly higher proliferation index than those expressing wild-type KLF5 or KLF5$^{KQ}$ (Fig. 2g, h), even though KLF5 tumors and KLF5$^{KQ}$ tumors caused more metastatic lesions than KLF5$^{KR}$ tumors (Fig. 2a–d). Therefore, more extensive metastatic tumor formation and more severe bone lesions caused by the acetylation of KLF5 at K369 are caused by reasons other than tumor cell proliferation rate.

**KLF5$^{KQ}$ is less potent than KLF5 and KLF5$^{KR}$ in tumor growth promotion in a non-bone environment but still maintains a mesenchymal phenotype.** To further address the roles of KLF5 and its acetylation in autonomous cell proliferation rate and tumor growth in a non-bone environment, we assessed tumor sphere formation in vitro and subcutaneous tumor formation in vivo. Sphere formation and tumor formation were both minimal with PCa cells lacking KLF5, and were rescued by the restoration of KLF5 regardless of acetylation status (Fig. 3a–h). While expression of KLF5$^{KQ}$ induced both sphere-forming (Fig. 3b–d) and tumorigenic abilities (Fig. 3e–h), expression of KLF5 or KLF5$^{KR}$ caused more and larger spheres (Fig. 3b–d) and more rapid tumor growth (Fig. 3e–h). Rates of cell proliferation in these subcutaneous tumors, as indicated by IHC staining of Ki67 (Fig. 3i, j and Supplementary Fig. 5a, b), were consistent with tumor growth rates (Fig. 3e–h). Apoptotic signaling, as indicated by increased expression of cleaved caspase 3, was activated by loss of KLF5 but not by KLF5, KLF5$^{KR}$, or KLF5$^{KQ}$ (Fig. 3i, k and Supplementary Fig. 3a, c).

Different effects of KLF5$^{KQ}$ and KLF5$^{KR}$ on EMT were maintained in tumors, as KLF5$^{KQ}$ tumors showed a mesenchymal phenotype, while KLF5$^{KR}$ tumors showed an epithelial phenotype, as indicated by the patterns of E-cadherin and vimentin expression (Fig. 3l–n and Supplementary Fig. 5d–f). Notably, cells isolated from subcutaneous tumors maintained not only

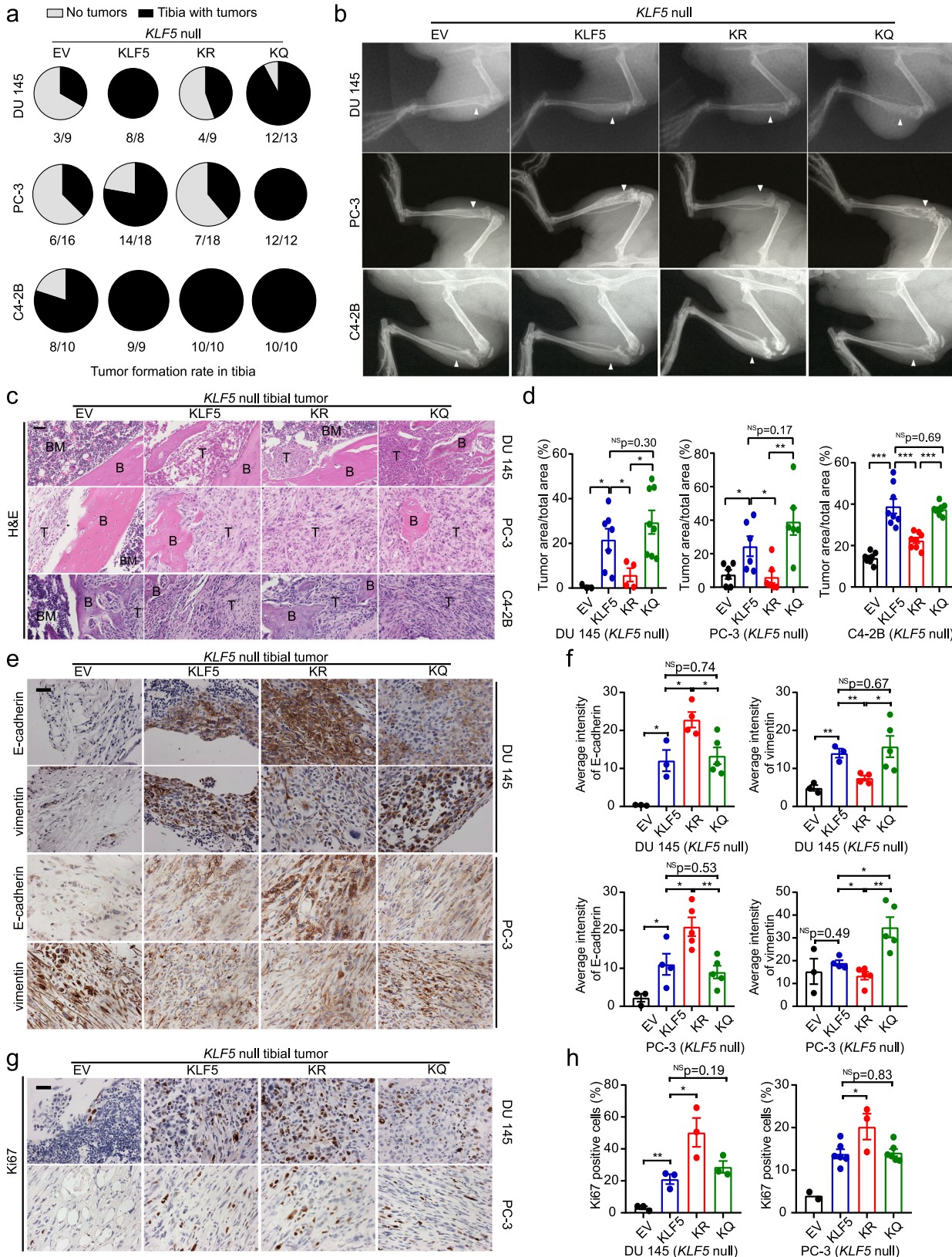

consistent EMT phenotypes during 2-D culture (Supplementary Fig. 5g) but also consistent sphere formation capabilities during two rounds of serial passaging (Supplementary Fig. 5h–n). These findings indicate that while KLF5 is indispensable for tumor formation, acetylation renders KLF5 less potent in tumor growth but able to maintain a mesenchymal morphology in tumors.

**Acetylation of KLF5 in tumor cells is required for osteoclast differentiation during bone metastasis.** In addition to tumor cell proliferation, bone metastatic growth of PCa cells also causes bone lesions via their interactions with the bone microenvironment including both osteoclasts and osteoblasts; and both osteolytic and osteoblastic lesions release factors that promote

**Fig. 2 Acetylation of KLF5 promotes cancer cell-induced bone metastatic lesions while maintaining the mesenchymal phenotype of cancer cells. a**, **b** Bone metastatic growth of PCa cells (DU 145, PC-3, and C4-2B) expressing different forms of KLF5, including tumor formation rate (**a**) and radiographs (**b**) at 5 weeks (DU 145 and PC-3) or 12 weeks (C4-2B) after tibial inoculation of cells. PCa cells with *KLF5* knockout (KLF5 null) were infected with lentiviruses to express empty vector (EV), wild-type KLF5 (KLF5), KLF5K369R (KR), and KLF5K369Q (KQ). White arrows point to bone lesion areas. **c** H&E staining of tibial tumor samples from PCa cells with indicated forms of KLF5. B trabecular bone regions, BM bone marrow regions, T tumor regions. **d** Statistical analysis of the images in **c** by calculating the ratio of tumor area to total sample area of DU 145 (left), PC-3 (middle), and C4-2B (right) cells with different forms of KLF5. For DU 145 cells, *n* = 3, 7, 4, and 8 tumors for EV, KLF5, KR, and KQ respectively. For PC-3 cells, *n* = 6 tumors for each group. For C4-2B cells, *n* = 8 tumors for each group. **e**, **f** IHC staining (**e**) and intensity quantification (**f**) of epithelial marker E-cadherin and mesenchymal marker vimentin in tibial tumors of DU 145 and PC-3 cells with different forms of KLF5. For DU 145 cells, *n* = 3, 3, 4, and 5 tumors for EV, KLF5, KR, and KQ respectively. For PC-3 cells, *n* = 3, 4, 5, and 5 tumors for EV, KLF5, KR, and KQ respectively. **g**, **h** IHC staining (**g**) and signal intensity quantification (**h**) of proliferation marker Ki67 in tibial tumors of DU 145 (Left) and PC-3 (right) cells. *n* = 3 tumors for each group of PC-3 cells. For DU 145 cells, *n* = 2, 6, 3, and 6 tumors for EV, KLF5, KR, and KQ respectively. One representative field per tumor was used for statistical analysis in **e**–**h**. Scale bar, 50 μm. In panels **d**, **f**, and **h**, data are shown in mean ± S.E.M. NS not significant; *$p < 0.05$; **$p < 0.01$ (two-tailed Student's *t* test). The Fiji software was used to quantify staining signal intensities in panels **f** and **h**. Source data are provided as a Source Data file.

---

metastatic growth of tumor cells[13]. We therefore tested whether acetylation of KLF5 induces bone metastatic lesions by causing osteoclastogenesis or differentiation of osteoclasts, which are multinucleated cells of hematopoietic origin responsible for the degradation of old bone matrix.

We first detected the occurrence of osteoclasts at the tumor-bone interface by staining tartrate-resistant acid phosphatase (TRAP), a marker for osteoclasts in bone, in tumor-bearing tibias (Fig. 4a, b). Intriguingly, more TRAP-positive (TRAP + ) cells indeed formed at the tumor-bone interfaces by DU 145, PC-3, and C4-2B tumor cells carrying either KLF5 or KLF5KQ, indicating that KLF5 acetylation in tumor cells promotes osteoclastogenesis. Occurrence of TRAP + osteoclasts in the KLF5 group was comparable to that in the KLF5KQ group (Fig. 4a, b), consistent with TGF-β inducing the acetylation of KLF5 and bone resorption releasing a large amount of TGF-β (Fig. 1). KLF5-null cancer cells, which do not have Ac-KLF5, also have less TRAP + osteoclasts (Fig. 4a, b).

We then tested in vitro whether acetylation of KLF5 indeed causes osteoclast differentiation. We co-cultured the mouse preosteoclast RAW264.7 cells with DU 145, PC-3 and C4-2B cells expressing different forms of KLF5, and measured the formation of osteoclasts indicated by positive staining of TRAP. Interruption of KLF5 acetylation at K369 suppressed osteoclast differentiation in vitro (Fig. 4c, d). Consistently, interruption of KLF5 acetylation also decreased the mRNA levels of multiple markers of osteoclast differentiation in co-cultured RAW264.7 cells, including *Trap*, *Nfatc1*, *Itgb3*, *c-Myc*, *Tm7sf4*, *Ctsk*, *Mmp9*, and *c-Src*, as detected by real-time qPCR (Fig. 4e). Therefore, acetylation of KLF5 in tumor cells is indeed required for inducing osteoclast differentiation.

We also tested the effects of conditioned media (CM) from cancer cells on osteoclast differentiation. CM from DU 145 cells with different forms of KLF5 were used to treat the preosteoclast RAW264.7 cells in the presence of 20 ng/ml Rankl (Fig. 4f, g), which effectively boosts the basal level of osteoclast differentiation (Supplementary Fig. 6a–c). Consistent with findings from the co-culture system, CM from cells expressing KLF5KR significantly reduced mature osteoclasts, as indicated by fewer TRAP + cells (Fig. 4f, g) and reduced expression of osteoclast markers (Fig. 4h), further indicating that interruption of KLF5 acetylation in PCa cells suppresses osteoclast differentiation.

Bone metastasis of PCa involves both osteoclasts and osteoblasts, so we also tested whether CM from PCa cells affects the genesis of osteoblasts. As expected, differentiation medium (DM), which contains differentiation-inducing ascorbic acid and β-glycerophosphate, successfully induces osteoblast differentiation, as indicated by Alizarin red S staining (Supplementary Fig. 6d–f) and expression changes of osteoblast differentiation

markers *Bglap*, *IL6*, *Osx*, and *Runx2* (Supplementary Fig. 6g)[47]. However, CM from DU 145 PCa cells with different forms of KLF5 did not affect osteoblast differentiation, either in normal or differentiation medium, as indicated by Alizarin red S staining (Supplementary Fig. 6h–j) and expression of osteoblast differentiation markers (Supplementary Fig. 6k). However, *IL6* level was significantly induced by both KLF5 and KLF5KQ but not by KLF5KR (Supplementary Fig. 6k). IL-6 is a cytokine released by osteoblasts to induce osteoclast differentiation[48], so IL-6 induced in osteoblasts by KLF5KQ-expressing tumor cells could be a paracrine signaling that mediates the effect of KLF5KQ on osteoclast differentiation.

**CXCR4 is transcriptionally induced by Ac-KLF5 to promote osteoclast differentiation.** KLF5 is a basic transcription factor that regulates multiple cellular processes via gene regulation[49], and its acetylation at K369 leads to transcriptional regulation of specific genes[38,40,41,50,51]. To identify which genes mediate the function of Ac-KLF5 in osteoclast differentiation, we performed RNA-Seq, ChIP-Seq, and bioinformatics analyses in KLF5-null PCa cells expressing *KLF5*, *KLF5KR*, and *KLF5KQ*. Expression of KLF5 in both DU 145 and PC-3 cells induced 294 and 62 genes respectively (Supplementary Fig. 7a, c). While many genes were similarly regulated by KLF5KQ and KLF5KR, 84 and 32 genes were up- and downregulated, respectively, only by KLF5KQ in DU 145 cells (Supplementary Fig. 7b and Supplementary data 1), and 71 and 35, respectively, in PC-3 cells (Supplementary Fig. 7d and Supplementary data 2). Overlapping genes between PC-3 and DU 145 cells were also observed for KLF5 (Supplementary Fig. 7e and Supplementary data 3) and KLF5KQ (Supplementary Fig. 7f and Supplementary data 4).

ChIP-Seq assay revealed 4007 binding regions for KLF5 in the promoters of its downstream genes in DU 145 cells (Supplementary Fig. 7g and Supplementary data 5). Although most of the binding regions were shared by KLF5KQ and KLF5KR, 424 peaks were specific to KLF5KQ and 11 to KLF5KR (Supplementary Fig. 7h and Supplementary data 5), suggesting that KLF5KQ has stronger binding affinities to promoter regions. Overlapping genes between ChIP-Seq and RNA-Seq analyses in DU 145 cells revealed that promoters of some genes induced by KLF5 (Supplementary Fig. 7i and Supplementary data 6) or KLF5KQ (Fig. 5a and Supplementary data 7) were bound by KLF5, identifying several directly targeted genes of KLF5 (Supplementary Fig. 7i) and KLF5KQ (Fig. 5a). Some genes with differential expression in RNA-Seq did not display promoter binding by KLF5 in ChIP-Seq, suggesting that they are indirectly regulated by KLF5 (Supplementary Fig. 7j and Supplementary data 8) and KLF5KQ (Supplementary Fig. 7k and Supplementary data 9).

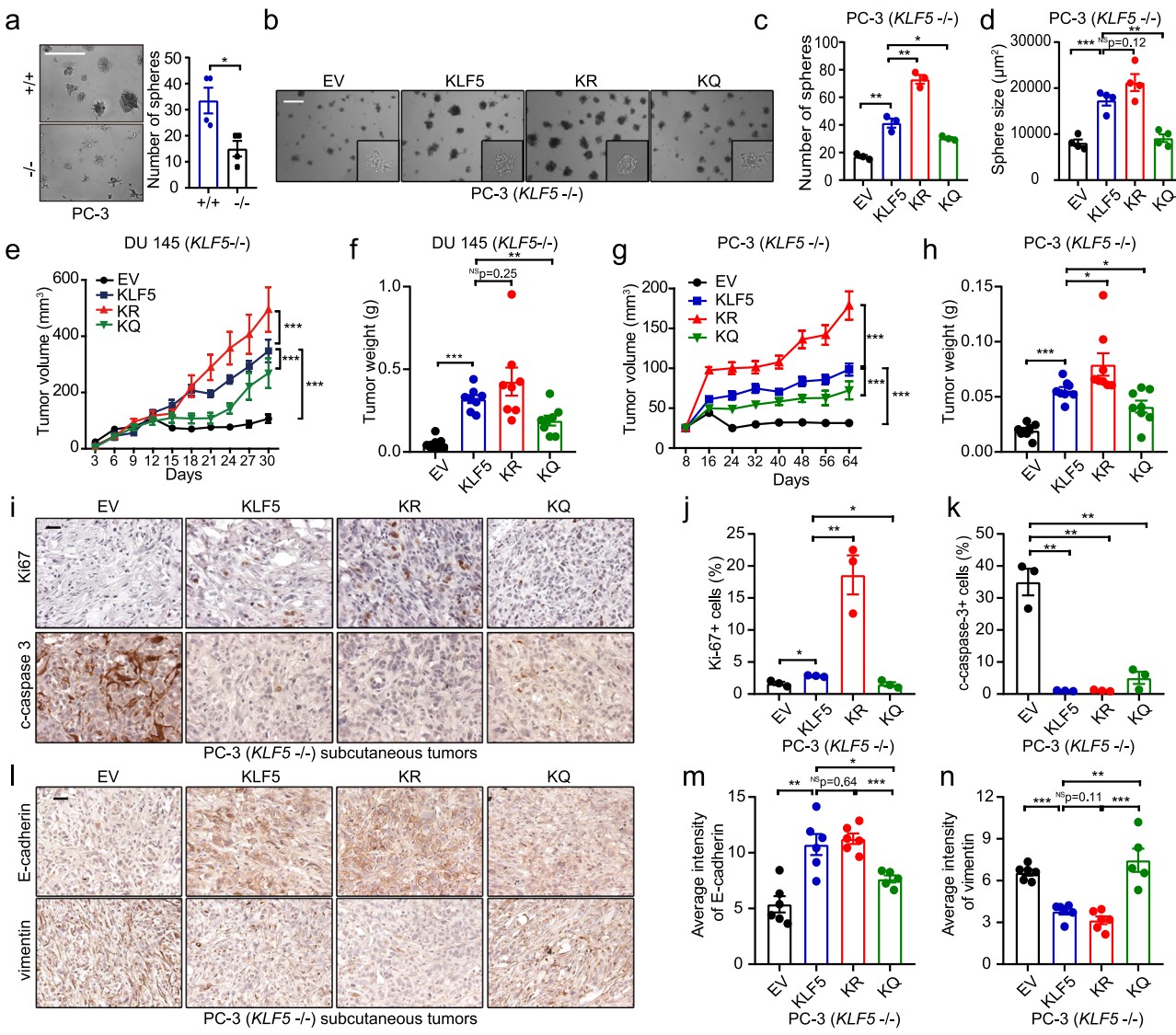

**Fig. 3 KLF5 is essential for subcutaneous tumor growth, but KLF5^KQ is weaker while KLF5^KR is stronger than KLF5 in their tumorigenic activities. a** Deletion of *KLF5* inhibited sphere formation in PC-3 cells. +/+, KLF5 wildtype; −/−, KLF5 null. Numbers of spheres in four different wells were analyzed. **b–d** Expression of wild-type *KLF5* (KLF5), KLF5^KR (KR), or KLF5^KQ (KQ) rescued sphere formation but with different efficiencies, as indicated by representative images (**b**), numbers (**c**), and sizes (**d**) of spheres. EV empty vector. Scale bars in **a** and **b**, 500 μm. Three different wells were analyzed for sphere numbers in **c**, and four randomly selected spheres were used for analyzing size in **d** for each condition. **e–h** Subcutaneous tumor growth of DU 145 (**e**, **f**) and PC-3 (**g**, **h**) cells with different forms of KLF5, as indicated by tumor volumes at different times (**e**, **g**) and tumor weights at excision (**f**, **h**). Each group had eight tumors. **i–k** Detection of Ki67 and cleaved caspase 3 by IHC staining (**i**) and quantitative analyses of Ki67 (**j**) and cleaved caspase 3 (c-caspase-3) (**k**) positive rates in tumor xenografts of PC-3 cells. *n* = 3 tumors for each group. **l–n** IHC staining (**l**) and quantification of average staining intensities (**m**, **n**) of epithelial markers E-cadherin and mesenchymal marker vimentin in PC-3 tumor xenografts. *n* = 6 tumors for EV, KLF5, and KR groups and *n* = 5 tumors for the KQ group. IHC staining intensities were quantified with the Fiji software. Scale bars in **i** and **l**, 50 μm. In panels **c**, **d**, **f**, **h**, **j**, **k**, **m**, **n**, data are shown in mean ± S.E.M. NS not significant; *$p < 0.05$; **$p < 0.01$; ***$p < 0.001$ (two-tailed Student's *t* test). In panels **e** and **g**, ***$p < 0.001$ (two-way ANOVA tests). Source data are provided as a Source Data file.

Considering data from all three analyses—RNA-Seq in both PC-3 and DU 145 cells and ChIP-Seq in DU 145 cells (Fig. 5b)—we identified five genes that were upregulated and two downregulated by KLF5^KQ across all analyses. A functional screening was then applied to these genes to identify those that mediate the function of KLF5 acetylation in osteoclast differentiation (Fig. 5c). For each of the 5 KLF5^KQ-upregulated genes, i.e., *LGR6*, *PCBP3*, *AXIN2*, *RETREG1*, and *CXCR4* (Fig. 5b), a mixture of shRNAs was expressed to knock down their expression in KLF5^KR and KLF5^KQ-expressing DU 145 cells; and the osteoclast differentiation assay was performed using co-cultures. Two genes, *LGR6* and *CXCR4*, stood out, as each of their knockdowns prevented the induction of

osteoclastogenesis by KLF5^KQ (Fig. 5c). CXCR4 and LGR6 are thus likely mediators of Ac-KLF5 in osteoclast differentiation.

Although the promoters of *CXCR4* and *LGR6* were bound by KLF5, as revealed by ChIP-Seq analysis, only the *CXCR4* promoter showed a differential binding between KLF5^KQ and KLF5^KR (Fig. 6f). We thus focused on CXCR4 as the downstream mediator of Ac-KLF5 in the induction of osteoclast differentiation. Blockage of CXCR4 signaling has been shown to inhibit PCa bone metastasis[52,53], but a role of CXCR4 in the communication between osteoclasts and tumor cells is still not clear in PCa. We found that knockdown of *CXCR4* by two different shRNAs A8 and A9 effectively inhibited osteoclast differentiation induced by

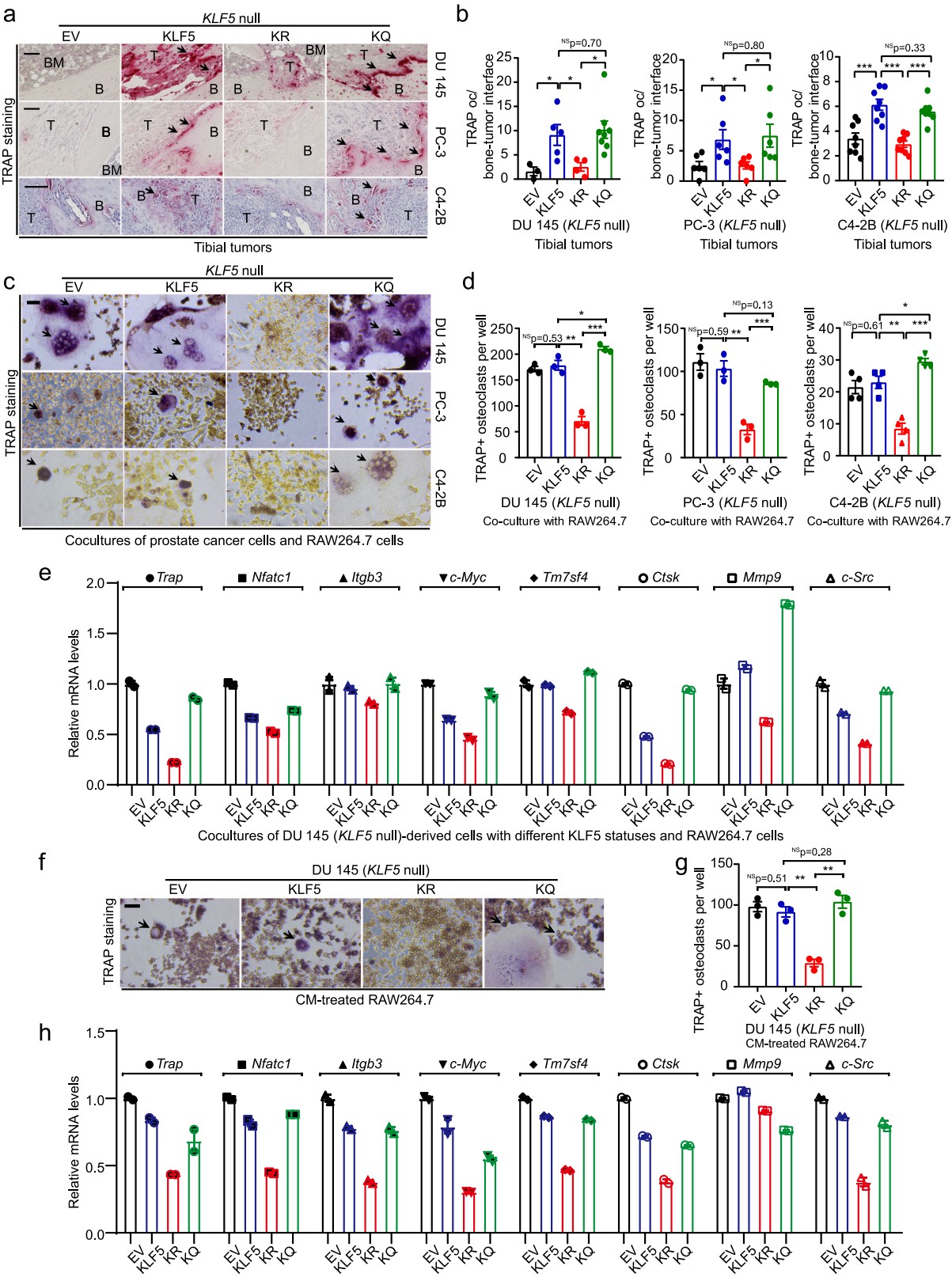

KLF5$^{KQ}$-expressing cells, as indicated by TRAP staining (Fig. 5d, e and Supplementary Fig. 8b) and real-time qPCR assay of markers for osteoclast differentiation (Fig. 5f). In addition, the promoting effect of KLF5$^{KQ}$ on osteoclast differentiation was enhanced by adding CXCL12, ligand of CXCR4, to the media; but was abolished by the inhibition of CXCR4 either via its inhibitor

AMD3100 or via the knockdown of *CXCR4*, as indicated by TRAP staining (Fig. 5g, h) and mRNA levels of *Trap* (Fig. 5i) and *Tm7sf4* (Fig. 5j). Therefore, induction of CXCR4 is essential for Ac-KLF5 to promote osteoclast differentiation.

We further characterized the transcriptional activation of *CXCR4* by KLF5$^{KQ}$ in PCa cells. Firstly, the expression of *CXCR4*

**Fig. 4 Acetylation of KLF5 in PCa cells promotes osteoclast differentiation. a, b** Staining of TRAP (**a**) and TRAP occurrence rates at the bone-tumor interface of bone samples bearing DU 145 (**b**, left), PC-3 (**b**, middle), and C4-2B (**b**, right) cancer cells with different acetylation statuses of KLF5. Black arrows in panel **a** indicate TRAP occurrence at the interface of bone and tumor areas (marked by B and T respectively). Scale bar in **a**, 100 μm. For DU 145 tibial tumors, EV, KLF5, KLF5$^{KR}$ (KR), and KLF5$^{KQ}$ (KQ) had 3, 5, 4, and 8 tumors respectively. For PC-3 tibial tumors, each group had six tumors. For C4-2B cells, $n = 8$ tumors for each group. **c, d** Differentiation of preosteoclast RAW264.7 cells into TRAP-positive osteoclasts after co-culturing with DU 145 (**d**, left), PC-3 (**d**, middle), and C4-2B (**d**, right) cells with different statuses of KLF5 acetylation, as indicated by TRAP staining (**c**, TRAP + multinucleated osteoclasts are marked by black arrows) and statistical analyses of TRAP + osteoclasts per well (**d**). $n = 3$ wells for each group in DU 145 and PC-3 cells. $n = 4$ wells for each group in C4-2B cells. **e** Expression of markers for osteoclast differentiation, including *Trap, Nfatc1, Itgb3, c-Myc, Tm7sf4, Ctsk, Mmp9,* and *c-Src*, by real-time qPCR using mouse-specific primers in the co-cultures of RAW264.7 mouse pre-osteoclasts and DU 145 cells with different forms of KLF5. **f, g** Staining of TRAP (**f**) and statistical analysis of TRAP + osteoclasts per well (**g**) in RAW264.7 pre-osteoclasts treated with CM from DU 145 cells with different forms of KLF5 for 6 days. Black arrows in **f** indicate TRAP + multinucleated osteoclast cells, and $n = 3$ wells for each group in **g**. **h** Detection of osteoclast differentiation markers in CM-treated RAW264.7 pre-osteoclasts by real-time qPCR. For both co-culture and CM treatment experiments, Rankl (20 ng/ml) was added to maintain a basal level of osteoclastogenesis. Scale bars in **c** and **f**, 50 μm. In panels **b**, **d** and **g**, data are shown in mean ± S. E.M. NS not significant; *$p < 0.05$; **$p < 0.01$; ***$p < 0.001$ (two-tailed Student's *t* test). Source data are provided as a Source Data file.

mRNA was shown by real-time qPCR to be upregulated by KLF5$^{KQ}$ in both DU 145 (Fig. 6a) and PC-3 (Fig. 6b) cells. Since CXCR4 is a membrane protein, we analyzed its distribution and expression level by flow cytometry in cultured cells. In both DU 145 (Fig. 6c) and PC-3 (Fig. 6d) cells, CXCR4 protein level was significantly increased by KLF5$^{KQ}$ (Supplementary Fig. 8a). IHC staining of CXCR4 in tibial tumors also demonstrated an increase in CXCR4 protein in KLF5 and KLF5$^{KQ}$-expressing tumors (Fig. 6e). Secondly, a unique region in the CXCR4 promoter was identified in ChIP-Seq analysis for its preferential binding by KLF5$^{KQ}$ (Fig. 6f), and the preferential binding was validated by ChIP-PCR (Fig. 6g). Furthermore, four putative KLF5-binding elements (KBEs) were detected in the differential binding region of the *CXCR4* promoter by using a web-based bioinformatics tool (oPROF), and luciferase reporter assay confirmed that KLF5$^{KQ}$ induced higher promoter activity than KLF5$^{KR}$, regardless of the number of covered KBEs (Fig. 6h). These findings indicate that induction of CXCR4 by Ac-KLF5 involves its preferential binding to specific sequences in the *CXCR4* promoter.

**Paracrine IL-11 mediates the promoting effect of the Ac-KLF5/CXCR4 axis on osteoclast differentiation.** The findings that CM from KLF5$^{KQ}$-expressing cells promoted osteoclast differentiation (Fig. 4f–h) and knockdown of *CXCR4* in the same cells eliminated this effect (Fig. 7a and Supplementary Fig. 8c) suggest that the Ac-KLF5/CXCR4 axis activates paracrine signaling to promote osteoclast differentiation. To define the underlying paracrine signaling, we analyzed a panel of cytokines involved in the tumor-bone interaction for KLF5$^{KQ}$-induced expression changes using real-time qPCR. Among the cytokines tested, the expression of *SHH, WNT5A, IL11,* and *IL6* was induced but that of *IL18* was reduced by KLF5$^{KQ}$ (Fig. 7b). Interestingly, knockdown of *CXCR4* prevented only the induction of *IL11* but not that of *SHH, IL6,* or *WNT5A* nor the reduction of *IL18* by KLF5$^{KQ}$ in co-cultures of DU 145 and RAW264.7 cells (Fig. 7c). Furthermore, a functional screening assay demonstrated that only knockdown of *IL11* significantly attenuated the promoting effect of KLF5$^{KQ}$ on osteoclast differentiation in the co-culture system (Fig. 7d). These findings suggest that *IL11* is a downstream target gene of the Ac-KLF5/CXCR4 axis that mediates the function of Ac-KLF5 in bone metastatic growth.

Furthermore, knockdown of *CXCR4* in KLF5$^{KQ}$-expressing DU 145 and PC-3 cells sharply decreased both the transcription of *IL11*, as detected by real-time qPCR (Fig. 7e, f), and the secretion of IL-11 protein, as detected by ELISA (Fig. 7g, h). ELISA also demonstrated that KLF5$^{KQ}$- and KLF5$^{KR}$-expressing cells secrete different amounts of IL-11, i.e., ~510 and 430 pg/ml for DU 145, and ~130 and 80 pg/ml for PC-3 cells, respectively (Fig. 7g, h). We added IL-11 to CM from KLF5$^{KR}$-expressing cells

(80 pg/ml for DU 145 and 50 pg/ml for PC-3) to obtain equal levels between KLF5$^{KQ}$- and KLF5$^{KR}$ groups in each cell line. IL-11 supplementation rescued the impaired osteoclast differentiation in KLF5$^{KR}$-expressing cells (Fig. 7i, j and Supplementary Fig. 8d, e). Similarly, attenuation of osteoclast differentiation by the knockdown of *CXCR4* in KLF5$^{KQ}$-expressing cells was rescued by adding IL-11 to the level without *CXCR4* knockdown (Fig. 7k, l and Supplementary Fig. 8d, e). Furthermore, knockdown of IL-11 not only abolished the induction of osteoclast differentiation by KLF5$^{KQ}$ (Supplementary Fig. 8f), it also eliminated the suppressive effects of CXCR4 knockdown on osteoclast differentiation (Supplementary Fig. 8g). Taken together with the reports that IL-11 binds to its receptors on osteoclasts or their precursors to regulate the formation and function of osteoclasts[54,55], these findings suggest that the Ac-KLF5/CXCR4 axis in tumor cells induces osteoclast differentiation by increasing IL-11 secretion.

We also noticed higher levels of *SHH* in KLF5$^{KQ}$-expressing cells (Fig. 7b). SHH released by tumor cells is able to act on osteoblasts to increase their production of IL-6, which in turn promotes osteoclast differentiation during bone metastasis[47]. Based on the observation that *IL6* mRNA level was also higher in MC3T3-E1 preosteoclasts co-cultured with KLF5$^{KQ}$-expressing tumor cells (Supplementary Fig. 6k), we confirmed by ELISA that secretion of IL-6 protein was indeed increased in the media of co-cultures (Supplementary Fig. 8h). Functionally, CM from the co-cultures enhanced osteoclast differentiation, to a greater degree than the sum of CM effects from MC3T3-E1 cells and cancer cells (Supplementary Fig. 8i). It is thus likely that SHH induced by KLF5$^{KQ}$ in tumor cells contributes to osteoclast differentiation by acting on osteoblasts to increase their release of IL-6.

**Pharmaceutical inhibition of CXCR4 sensitizes bone metastatic tumors to docetaxel.** At present, the taxanes docetaxel and cabazitaxel are the only chemotherapeutic agents that have a survival benefit for patients with mCRPC. However, virtually all mCRPCs eventually develop resistance[8] and patients with bone metastasis still have poor prognosis after docetaxel treatment[9]. Considering that Ac-KLF5 and CXCR4 plays a role in docetaxel resistance in some cancers[56–59]; and we have observed that *CXCR4* is transcriptionally upregulated by KLF5$^{KQ}$ (Fig. 6), the Ac-KLF5/CXCR4 axis plays a necessary role in tumor-induced osteoclast differentiation (Fig. 5), and Ac-KLF5 promoted bone metastatic growth (Fig. 2), it is plausible that CXCR4 expression induced by KLF5$^{KQ}$ mediates the resistance of KLF5$^{KQ}$-expressing bone metastases to docetaxel.

We therefore tested whether inhibition of CXCR4 enhances the therapeutic effect of docetaxel using the tibial injection mouse model to mimic the PCa patients who have developed bone

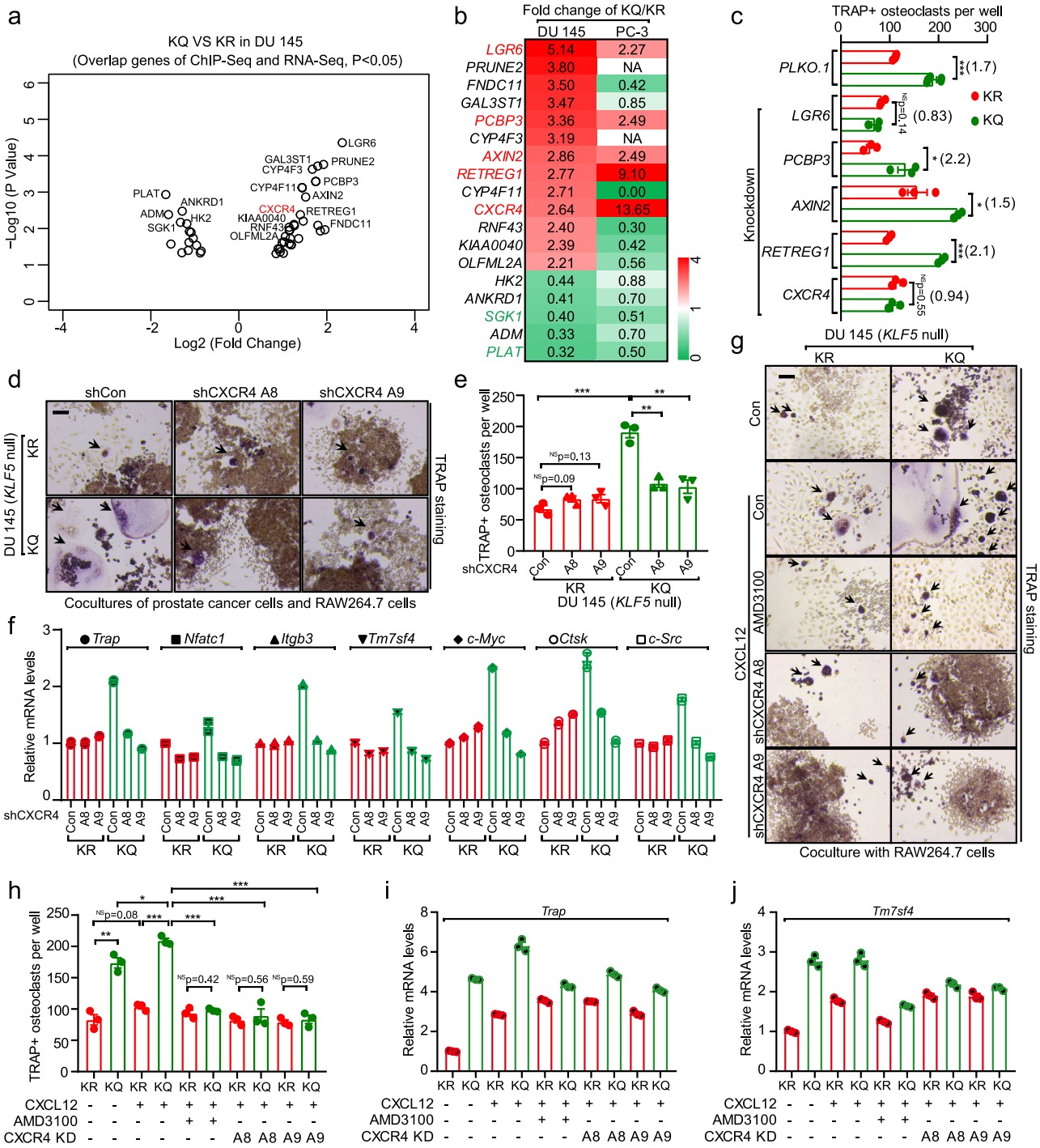

metastasis. In mice bearing tibial tumors of parental PC-3 cells, in which Ac-KLF5 was induced by enriched TGF-β in the bone (Fig. 1a–f), docetaxel alone slightly suppressed tibial tumor growth, as indicated by the reduced tumor size (Fig. 8a, b); and addition of CXCR4 inhibitor AMD3100 (plerixafor) significantly sensitized the tibial tumors to docetaxel, as demonstrated by alleviated bone lesions and reduced tumor area (Fig. 8a, b). The coefficient of drug interaction (CDI) of docetaxel and AMD3100 was 0.54 (<0.7), suggesting that combined docetaxel and AMD3100 treatment had a synergistic effect on PCa bone metastasis. As expected, inhibition of CXCR4 by AMD3100 suppressed osteoclast differentiation in tibial tumors (Fig. 8a, c). For subcutaneous tumors, however, AMD3100 failed

to sensitize tumors to docetaxel, although docetaxel alone showed a suppressive effect on tumor growth (Supplementary Fig. 9a, b). Subcutaneous tumors expressed less Ac-KLF5 than their tibial counterparts (Fig. 1a–f), which could explain the differences between tibial and subcutaneous tumors.

To further confirm whether Ac-KLF5 is essential for the synergistic effect of docetaxel and AMD3100 on bone metastasis of PCa cells, we administered the CXCR4 inhibitor AMD3100 to mice carrying tibial tumors of KLF5$^{KQ}$ or KLF5$^{KR}$, with or without docetaxel treatment, and analyzed metastatic tumor growth by measuring alleviated bone lesions with X-ray radiographs (Supplementary Fig. 9c) and tumor areas (Fig. 8d, e). While docetaxel alone inhibited metastatic growth of KLF5$^{KR}$

**Fig. 5 Identification of CXCR4 as a functional effector of acetylated KLF5 in the induction of osteoclast differentiation. a** Differentially expressed genes between KLF5$^{KQ}$- expressing (KQ) and KLF5$^{KR}$-expressing (KR) DU 145 cells, as identified by ChIP-Seq and RNA-Seq analyses and illustrated by a volcano diagram according to their *p*-values and fold changes in RNA-Seq. Circles indicate genes whose promoter regions (−2500 to +500) had binding peaks in the ChIP-Seq assay. Details are available in Supplementary data 7. **b** Heatmap of genes in panel **a** with fold changes between KQ and KR in both DU 145 and PC-3 cells, as revealed by RNA-Seq analysis. Each rectangle is colored and the intensity is defined by the fold change of the gene expression levels in KQ versus KR cells. Red and green indicate upregulation and downregulation, respectively. Gene names in red and green indicate genes that are upregulated and downregulated, respectively, by KQ in both cell lines. **c** Functional screening for genes that mediate osteoclast differentiation induced by KLF5$^{KQ}$ (KQ)-expressing cells. Each of the KQ-upregulated genes from panel **b** was knocked down by a mixture of 2 or 3 shRNAs, each of which was confirmed to efficiently knock down its target gene, in both KQ and KR cells, and then co-cultured with RAW264.7 pre-osteoclasts. TRAP was stained to measure osteoclast differentiation. Numbers in parentheses indicate fold changes of TRAP + osteoclast numbers between KQ and KR. $n = 5$ wells for control shRNA *PLKO.1* and $n = 3$ wells for other shRNAs. **d, e** TRAP staining (**d**) and statistical analyses of TRAP + osteoclasts (**e**) in co-cultures of KR- or KQ-expressing DU 145 cells with RAW264.7 pre-osteoclasts in the presence of 20 ng/ml Rankl for 6 days. Black arrows indicate TRAP + multinucleated osteoclasts. A8 and A9 are two different CXCR4 shRNAs. $n = 3$ wells per group in **e**. **f** Knockdown of *CXCR4* selectively suppresses the expression of markers for osteoclast differentiation in KQ cells, as detected by real-time qPCR. **g, h** TRAP staining (**g**) and statistical analyses of TRAP + osteoclasts (**h**) in co-cultures of RAW264.7 and KR- or KQ-expressing DU 145 cells with inhibition of CXCR4 by either the AMD3100 inhibitor (500 ng/ml) or shRNAs (A8 and A9). The CXCR4 agonist ligand CXCL12 (100 ng/ml) was added as indicated. $n = 3$ wells per group in **h**. **i, j** Expression of osteoclast differentiation markers *Trap* (**i**) and *Tm7sf4* (**j**) was detected by real-time qPCR in co-cultures from panel **g**. Scale bar, 100 μm. In panels **c**, **e** and **h**, data are shown in mean ± S.E.M. NS not significant; *$p < 0.05$; **$p < 0.01$; ***$p < 0.001$ (two-tailed Student's *t* test). Source data are provided as a Source Data file.

tumors but had no significant effect on that of KLF5$^{KQ}$ tumors (Fig. 8d, e), AMD3100 alone suppressed metastatic growth of KLF5$^{KQ}$ tumors but not that of KLF5$^{KR}$ tumors (Fig. 8d, e). These results are consistent with the in vitro assay in which inhibition of CXCR4 selectively suppressed osteoclast differentiation induced by KLF5$^{KQ}$ tumor cells but not that by KLF5$^{KR}$ tumor cells (Fig. 5g, h). Interestingly, while AMD3100 treatment did not enhance the suppressive effect of docetaxel in KLF5$^{KR}$ tumors, combined treatment clearly had a synergistic effect in KLF5$^{KQ}$ tumors, because the CDI of docetaxel and AMD3100 was <0.7 in KLF5$^{KQ}$ tumors (0.59 for PC-3 and 0.40 for DU 145; Fig. 8d, e). No synergistic effect was detected in KLF5$^{KR}$ tumors, as the CDI was 0.97 for DU 145 tumors and 0.86 for PC-3 tumors. Therefore, inhibition of CXCR4 sensitizes KLF5$^{KQ}$-expressing tibial tumors to docetaxel.

We also analyzed the occurrence of TRAP at the tumor-bone interface to determine the effects of combined docetaxel and AMD3100 treatment on osteoclast differentiation in PCa bone metastasis (Fig. 8d, f). Consistent with the bone metastatic growth data, AMD3100 alone significantly decreased osteoclast occurrence in the KLF5$^{KQ}$ group but not in the KLF5$^{KR}$ group, as indicated by TRAP staining and quantitative analysis of TRAP + cells at the tumor-bone interface (Fig. 8d, f). On the other hand, docetaxel alone did not show a significant inhibitory effect in either the KLF5$^{KR}$ or KLF5$^{KQ}$ group (Fig. 8d, f); and combined AMD3100 and docetaxel treatment did not significantly enhance the effect of AMD3100. In addition to further supporting a role of CXCR4 in Ac-KLF5-mediated bone lesions, these findings suggest that combined docetaxel and AMD3100 treatment could be an effective therapy for patients with bone metastasis of PCa. We also detected the in vivo expression levels of IL-11 in tibial tumors by IHC staining and found that inhibition of CXCR4 by AMD3100 selectively suppressed IL-11 expression levels in KLF5$^{KQ}$ tibial tumors (Supplementary Fig. 9d, e), which further supports the conclusion that IL-11 is a paracrine signaling at the downstream of Ac-KLF5/CXCR4 axis in mediating osteoclast differentiation.

We also tested whether AMD3100 treatment only sensitizes KLF5$^{KQ}$ but not KLF5$^{KR}$ tumors to docetaxel using the subcutaneous xenograft model. In both DU 145 and PC-3 cells (Supplementary Fig. 9f–i), inhibition of CXCR4 by AMD3100 selectively sensitized KLF5$^{KQ}$-expressing subcutaneous tumors to docetaxel but failed to do so in KLF5$^{KR}$ tumors. Therefore, AMD3100-induced sensitization of tumors to the therapeutic

effect of docetaxel likely also occurs in localized tumors as long as the tumors have active Ac-KLF5/CXCR4 signaling.

**Acetylation of KLF5 commonly occurs and correlates with CXCR4 in bone metastases of PCa patients.** The expression of Ac-KLF5 was detected by IHC staining in samples from PCa patients, including normal tissues, hyperplasia, and localized tumors from prostates, and metastases in visceral organs and bone (Fig. 9a, b and Supplementary Fig. 10a). While Ac-KLF5 staining intensity was strong in basal cells and stroma of normal prostates, it was much weaker in luminal cells (Fig. 9a). The overall staining intensity was relatively lower in hyperplasia but was increased in malignant tissues (Fig. 9a, b). Impressively, the majority of the 51 bone metastases had more intense staining of Ac-KLF5 than the majorities of localized tumors and visceral metastases (Fig. 9a, b), which is consistent with an enriched TGF-β microenvironment in the bone with metastases (Fig. 1). The ratios of Ac-KLF5-positive cells varied among different sites of bone metastases and visceral metastases (Supplementary Fig. 10b, c). Therefore, acetylation of KLF5 commonly occurs in bone metastases of PCa. CXCR4 expression was also detected in the same set of bone metastases using IHC staining, and the expression of Ac-KLF5 and that of CXCR4 were positively correlated in bone metastases of PCa (Fig. 9c, d).

## Discussion

Focusing on PCa bone metastasis, which occurs in the majority of CRPCs and always develops resistance to chemotherapy, here we report a mechanism for the development of chemoresistant bone metastasis in PCa progression, i.e., heavy acetylation of the KLF5 transcription factor in the bone microenvironment causes bone metastatic lesions by activating the CXCL12/CXCR4 chemokine axis and other paracrine signaling pathways such as those of IL-11 and SHH (Fig. 9e). This mechanism can potentially impact the detection and treatment of PCa bone metastasis.

We provide multiple lines of evidence for the establishment of acetylated KLF5 (Ac-KLF5) as a potent player that causes bone metastatic lesions in PCa cells. For example, tibial injection of PCa cells expressing the wildtype KLF5 or the KLF5$^{KQ}$ mutant, which mimics Ac-KLF5[60,61], caused significant bone metastatic lesions, that of the acetylation-deficient KLF5$^{KR}$ mutant did not (Fig. 2a–d). In addition, a small proportion of localized human PCa tissues or visceral metastases expressed Ac-KLF5 but the

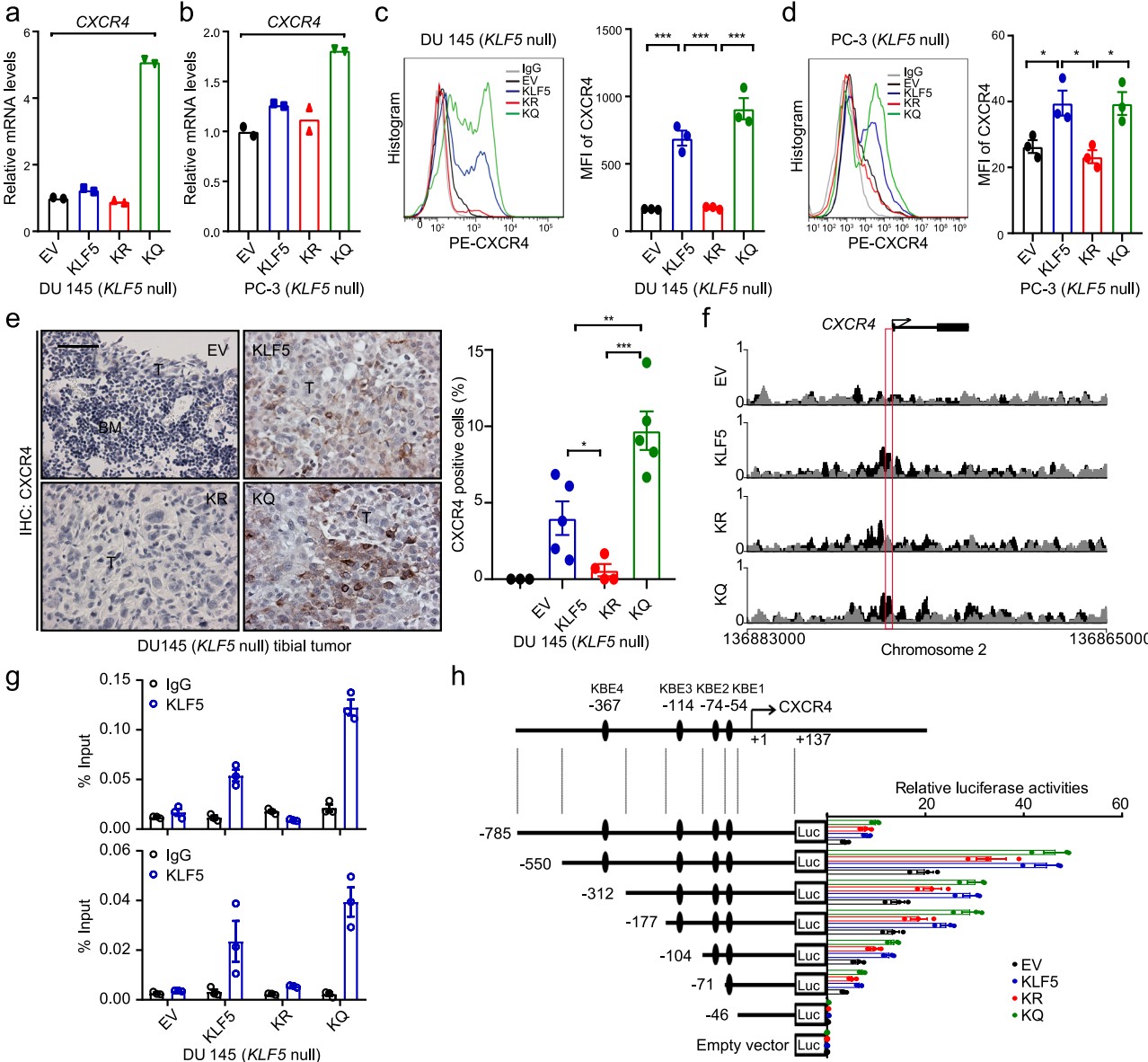

**Fig. 6 Acetylation of KLF5 activates the transcription of *CXCR4* in PCa cells. a–d** Expression of CXCR4, as detected by real-time qPCR (**a, b**) and flow cytometry (**c, d**), in DU 145 cells (**a, c**) and PC-3 cells (**b, d**) expressing different forms of KLF5 in in vitro 2-dimentiomal cultures. MFI mean fluorescent intensity. Experiments were performed in duplicate for real-time qPCR (**a, b**) and in triplicate for flow cytometry (**c, d**). See more details of panels **c** and **d** in Supplementary Fig. 8a. **e** IHC staining of CXCR4 in tumor-bearing bone samples. Rates of CXCR4-positive cells are shown at the right. BM bone marrow region, T tumor region. Scale bar, 50 μm. $n = 3, 5, 4$, and 5 tumors for EV, KLF5, KR, and KQ groups respectively. **f, g** A region in the *CXCR4* promoter, indicated by a red box, is specifically bound by KLF5$^{KQ}$ (KQ) but not by KLF5$^{KR}$ (KR), as demonstrated by ChIP-Seq analysis (**f**) and validated by ChIP-PCR with two different pairs of primers targeting the region (**g**). ChIP-PCR was performed in triplicate using in vitro two-dimentiomal cultures. **h** KQ is more potent than KR in the activation of *CXCR4* promoter, as revealed by promoter-luciferase reporter assay of different *CXCR4* promoter regions. Black ellipse, putative KLF5 binding motifs (KB) predicted by the oPROF web-based software. +1, transcription start site. Promoter-luciferase reporter assay was performed in triplicate. In panels **c–e**, data are shown in mean ± S.E.M. *$p < 0.05$; **$p < 0.01$; ***$p < 0.001$ (two-tailed Student's $t$ test). Source data are provided as a Source Data file.

majority of bone metastases did (Fig. 9a, b), which parallels with lower levels of TGF-β and Ac-KLF5 in subcutaneous tumors and higher levels of TGF-β[15,62] and Ac-KLF5 in tibial tumors (Fig. 1a–f), and TGF-β induces the acetylation of KLF5[38–40].

PCa cells expressing Ac-KLF5 also caused osteoclastogenesis, as indicated by a dramatic increase in TRAP-positive multi-nucleated osteoclasts in tibias (Fig. 4a, b) and TRAP + cells in the RAW264.7 in vitro model of osteoclastogenesis (Fig. 4c–h). These findings further indicate a role of Ac-KLF5 in metastatic growth of PCa in the bone, because both bone destruction and new bone

formation occur during bone metastasis, osteoclastogenesis is a crucial event that may precede or occur simultaneously with osteoblastogenesis[63], and tumor cells not only induce osteoclas-togenesis but also interact with osteoblasts and osteoclasts to induce bone metastasis via the activation of multiple paracrine factors[64].

Induction of osteoclastogenesis by Ac-KLF5-expressing tumor cells also indicates that Ac-KLF5 activates paracrine signaling in tumor cells to mediate tumor cells' interaction with osteoclasts or osteoblasts, which also provides a mechanism for how Ac-KLF5

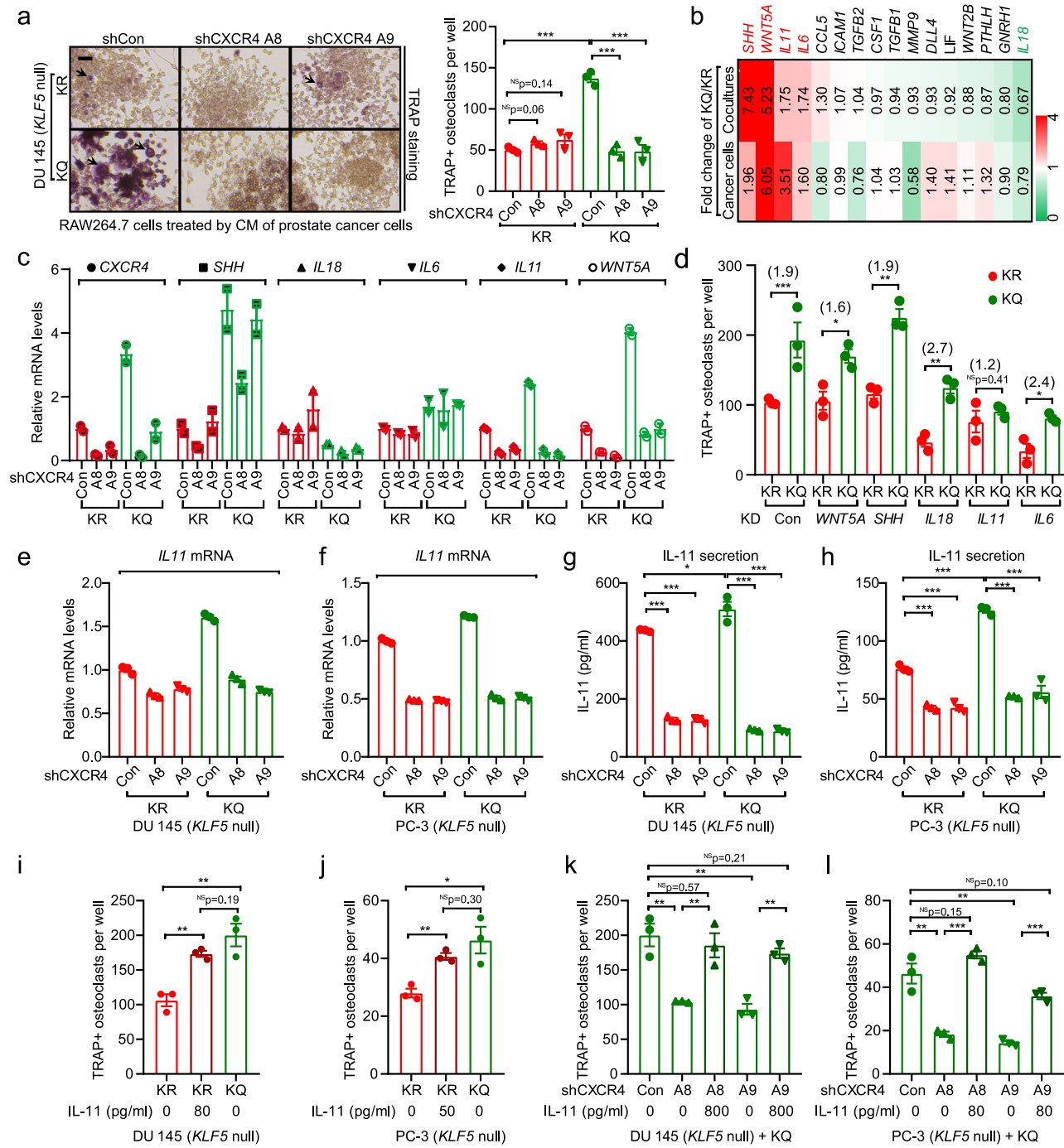

causes bone metastatic lesions. This was directly supported by the finding that CM from KLF5$^{KQ}$-expressing cells promoted osteoclast differentiation (Fig. 4f–h) while inhibition of CXCR4 in the same cells eliminated this effect (Fig. 7a and Supplementary Fig. 8c). Indeed, Ac-KLF5 transactivates the CXCL12/CXCR4 chemokine axis, IL-11 cytokine signaling, and likely other paracrine molecules in tumor cells, as revealed by RNA-Seq, ChIP-Seq, and related analyses (Fig. 5a, b and Supplementary Fig. 7). For example, the CXCR4 promoter had a unique sequence that was bound only by KLF5$^{KQ}$ but not by KLF5$^{KR}$ (Fig. 6f), and CXCR4 was indeed a direct transcriptional target gene of KLF5$^{KQ}$ in PCa cells (Fig. 6). KLF5 is a transcription factor that regulates multiple cellular processes via gene regulation[49]; and Ac-KLF5 forms a transcriptional complex that is different from that of

deacetylated KLF5, altering KLF5 functions in gene regulation and cell proliferation[38,40,41,50,51].

Tumor cells shed into the circulating system colonize in different organs to cause metastasis, and those colonized in the bone often interact with both osteoblasts and osteoclasts to cause metastatic lesions to the bone[65–67]. It appears that in the bone, KLF5 is more often acetylated due to the enrichment of TGF-β, and Ac-KLF5 activates multiple paracrine signaling pathways to induce osteoclast differentiation and bone metastatic lesions.

CXCR4, a G-protein-coupled receptor for the CXCL12 homeostatic chemokine promoting cell trafficking including the colonization of hematopoietic stem cells in bone marrow[67], the homing of tumor cell to organs with high levels of CXCL12 (e.g. lymph nodes, lungs, liver, and bone marrow), and bone

**Fig. 7 Acetylation of KLF5 activates *CXCR4* to promote osteoclastogenesis by increasing IL-11 secretion. a** Knockdown of *CXCR4* attenuates KLF5KQ-induced osteoclastogenesis via paracrine signaling. Preosteoclast RAW264.7 cells were treated with CM from KLF5KR (KR)- or KLF5KQ (KQ)-expressing DU 145 cells with or without CXCR4 knockdown, and TRAP was stained to indicate osteoclast differentiation. A8 and A9 are two different CXCR4 shRNAs. Scale bar, 100 μm. Black arrows indicate TRAP + multinucleated osteoclasts. *n* = 3 wells for statistical analysis. **b** Screening for paracrine factors upregulated by KQ but not by KR using real-time qPCR in either KQ/KR-expressing DU 145 cells or their co-cultures with RAW264.7 cells. Each rectangle is colored and the intensity is defined by the fold change of the gene expression levels in KQ versus KR cells. Red and green indicate upregulation and downregulation, respectively. **c** Expression changes of *IL11* but not the other 4 KQ-regulated cytokines, including *SHH, IL18, IL6*, and *Wnt5A* respond to *CXCR4* knockdown, as revealed by real-time qPCR in co-cultures of KQ- or KR-expressing DU 145 cells with RAW264.7 pre-osteoclasts. **d** Knockdown of *IL11* but not the other four KQ-upregulated cytokines attenuated osteoclastogenesis promoted by KQ, as indicated by TRAP staining in co-cultures of RAW264.7 cells and KQ- or KR-expressing PCa cells with the knockdown of different cytokines. PLKO.1 is the empty vector control for knockdown. Numbers in parentheses indicate fold changes in TRAP + osteoclasts between KQ and KR groups. *n* = 3 wells per group for statistical analysis. **e–h** Detection of IL-11 expression by real-time qPCR (**e, f**) and ELISA (**g, h**) in KR- or KQ-expressing DU 145 (**e, g**) and PC-3 (**f, h**) cells with or without *CXCR4* knockdown. Experiments were performed in triplicate. **i, j** Addition of IL-11 rescued the decrease in osteoclast differentiation of RAW264.7 cells by the CM from DU 145 (**i**) and PC-3 (**j**) cells that express KR. **k, l** Addition of IL-11 rescued the decrease in osteoclast differentiation of RAW264.7 cells from DU 145 (**k**) and PC-3 (**l**) cells that express KQ with the knockdown of *CXCR4*. *n* = 3 wells for statistical analysis. In panels **a, d, g–l**, data are shown in mean ± S.E.M. NS not significant; *\*p* < 0.05; *\*\*p* < 0.01; *\*\*\*p* < 0.001 (two-tailed Student's *t* test). Source data are provided as a Source Data file.

metastasis in PCa[52,53], indeed mediates the function of Ac-KLF5 in osteoclastogenesis and bone metastaticlesion. For example, inhibition of CXCR4 by either gene silencing or a pharmaceutical inhibitor prevented KLF5KQ from inducing osteoclast differentiation (Figs. 5g, h and 8d, f) and causing bone metastatic lesions (Fig. 8d, e). It is worth noting that tumor cells injected into tibias still required CXCR4 for Ac-KLF5 to induce bone metastatic lesions (Fig. 8); and CXCR4 was also necessary for the CM from Ac-KLF5 expressing cells to induce osteoclast differentiation (Figs. 4g and 7a), which suggests that CXCR4 might also have a role in paracrine signaling not necessarily related bone homing. Indeed, among several cytokines that regulate the tumor-bone interaction and were upregulated significantly more by KLF5KQ than by KLF5KR (*SHH, WNT5A, IL11*, and *IL6*; Fig. 7b), the induction of IL-11 was dependent on CXCR4 expression (Fig. 7e–h). In addition, IL11 indeed played a role in the promoting effect of KLF5KQ on osteoclast differentiation (Fig. 7d, i–l). It has been shown that IL-11 binds to its receptors on osteoclasts or their precursors to regulate the formation and function of osteoclasts and bone resorption[54,55,68].

In addition to IL11, several other paracrine molecules, including SHH, WNT5A, and SHH-induced osteoblast-secreted IL-6 (Fig. 7c), were also upregulated by KLF5KQ, even though their upregulation may or may not involve direct transcriptional regulation by Ac-KLF5 or depend on the CXCL12/CXCR4 axis in PCa cells. Such molecules could also mediate the effect of Ac-KLF5 on osteoclast differentiation and bone metastatic lesions. SHH could be one such molecule, as it has been demonstrated that SHH secreted by tumor cells acts on osteoblasts, which in turn release IL-6, to promote osteoclast differentiation[47]. In this study, although knockdown of *SHH* in KLF5KQ cells did not show an obvious effect on the differentiation of RAW264.7 pre-osteoclasts (Fig. 7d), co-cultures of KLF5- and KLF5KQ-expressing tumor cells with MC3T3-E1 preosteoblasts increased IL-6 expression in MC3T3-E1 cells, and CM from the co-cultures enhanced the differentiation of RAW264.7 preosteoclasts (Supplementary Figs. 6k and 8h, i). LGR6 could be another one, as it was induced by Ac-KLF5, and its induction was also involved in Ac-KLF5-promoted osteoclast differentiation in vitro (Fig. 5b, c).

TGF-β is a master regulator of EMT[69], a morphological transformation for tumor cells to adapt to specific microenvironments[70], and our findings indicate that it is the TGF-β/Ac-KLF5 axis rather than TGF-β alone that functions in EMT. For example, interruption of KLF5 acetylation prevented TGF-β from inducing EMT (Supplementary Fig. 2), and a role of Ac-KLF5 in EMT was also detected in both subcutaneous and tibial tumors (Figs. 2e, f and 3l–n, and Supplementary Figs. 3k–m

and 4) as well as in primary cultures of the subcutaneous tumors (Supplementary Fig. 5g). In addition, both blockage of TGF-β signaling and interruption of KLF5 acetylation reversed the EMT phenotypes of tibial tumors (Fig. 2e, f and Supplementary Fig. 1k). EMT features are associated with both bone metastases[22] and chemoresistance of PCa[71], and such associations were cleared detected in this study in the forms of KLF5KQ and KLF5KR mutants.

Transition between epithelial and mesenchymal morphologies upon changes in KLF5's acetylation status in cancer cells could have important implications. During tumor progression, plasticity enables cancer cells to change their morphology and behavior in order to survive stresses and to reverse these changes in order to continue their propagation. Our findings of Ac-KLF5 being essential for the mesenchymal phenotype (Figs. 2e and 3l, and Supplementary Figs. 2 and 4), tumorigenicity (Fig. 3), bone metastatic lesions (Fig. 2), and chemoresistance (Fig. 8 and Supplementary Fig. 9) establish KLF5 acetylation as a mechanism through which PCa cells change to survive stresses (i.e., by becoming mesenchymal and slowly proliferating). Posttranslational modifications of proteins, including the acetylation of KLF5, are reversible, which implies that when a stress is withdrawn, mesenchymal cancer cells could become epithelial and rapidly proliferate, causing tumor recurrence and faster growth. Taken together, it appears that the acetylation status of KLF5 switches PCa cells between drastically different states. One is the epithelial state induced and maintained by deacetylated KLF5, in which cells are more rapid in proliferation and tumor formation, ineffective in the induction of bone metastatic lesions, and more sensitive to the therapeutic effect of docetaxel. The other is the mesenchymal state maintained by Ac-KLF5, which makes cells slower in cell proliferation and tumor growth, potent in the induction of bone metastatic lesions, and resistant to docetaxel. KLF5-regulated cell plasticity is likely a crucial mechanism for cancer cells to survive stresses from tumor microenvironments, including the bone microenvironment and chemotherapy-mediated stresses.

TGF-β and Ac-KLF5 form a signaling axis to function, as TGF-β induces the acetylation of KLF5 and the subsequent formation of transcriptional complexes to regulate gene transcription and cell proliferation[38–41]. This is further supported by this study, as tumors in the tibia, which has more abundant TGF-β[72,73], clearly had more extensive KLF5 acetylation than subcutaneous tumors (Fig. 1a–f) and blockade of TGF-β signaling by the TGFBR1 inhibitor SD-208 eliminated KLF5 acetylation (Fig. 1a–f). Considering that TGF-β induces the acetylation of KLF5 to form a signaling axis in the regulation of gene transcription, cell

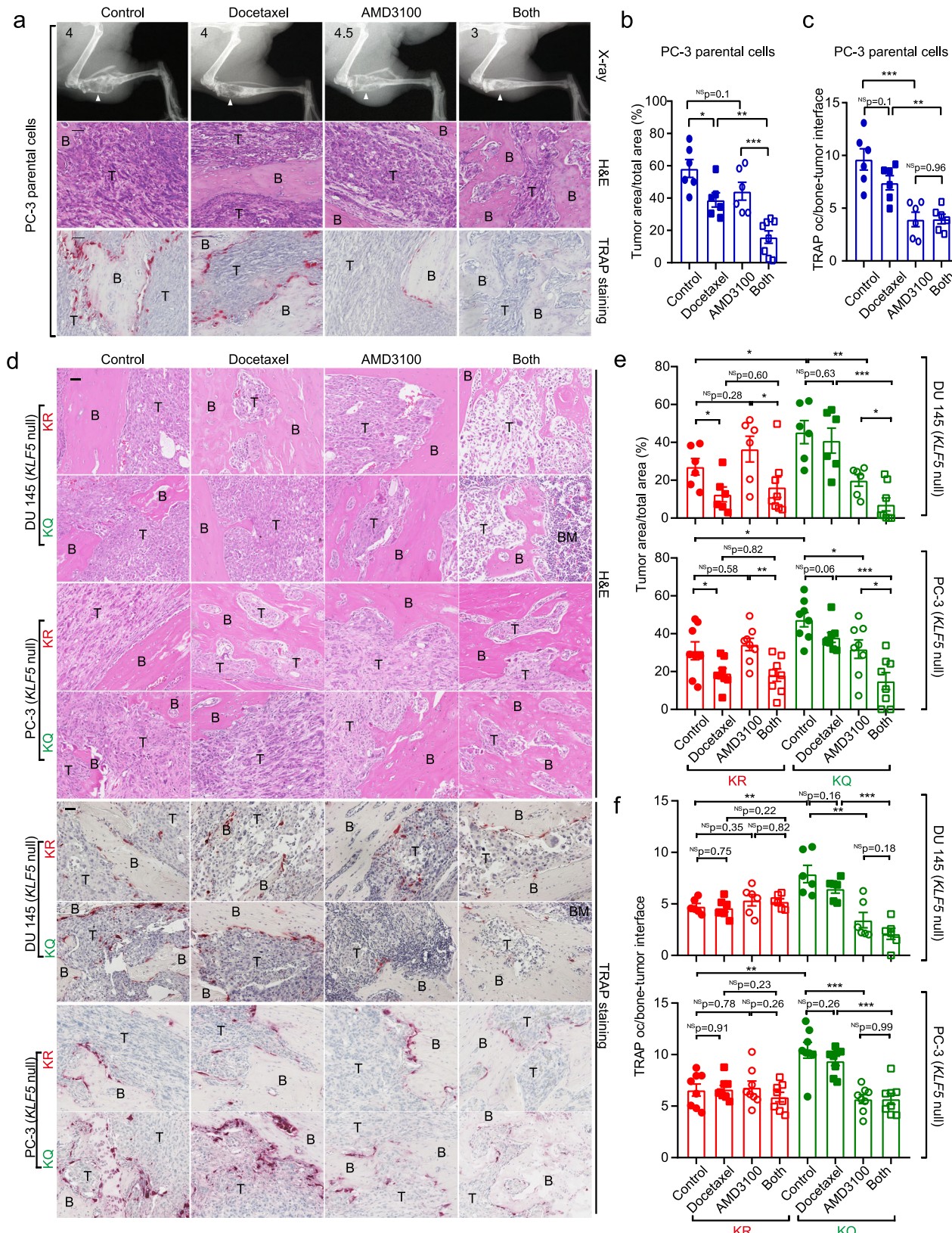

proliferation, EMT (see discussion above), osteoclast differentiation, and bone metastatic lesions[38–40] (Figs. 1, 2, and 5–8), we conclude that acetylation of KLF5 is a fundamental mechanism for TGF-β to function, and the TGF-β/Ac-KLF5 axis is an important signaling axis in the regulation of multiple biological processes in epithelial cells.

Ac-KLF5 functions as an effector of TGF-β signaling in bone metastasis. TGF-β is a potent promoter of bone metastasis, and inhibition of TGF-β signaling has been demonstrated to suppress bone metastasis of PCa[74,75]. During PCa bone metastasis, TGF-β acts on tumor cells to induce the secretion of paracrine signaling molecules to support their survival and growth; and some of them

**Fig. 8 Inhibition of CXCR4 enhances the therapeutic effect of docetaxel on Ac-KLF5-induced bone metastatic lesions. a** X-ray radiographs (up), H&E staining (middle), and TRAP staining (down) of tibias at 40 days after inoculation of PC-3 parental cells into the tibias. White arrows point to areas of bone lesions. Numbers at the top-left corner of X-ray radiographs are the Average Bone Lesion Score (ABLS) based on the degree of osteolysis. **b** The ratio of tumor area to total sample area of PC-3 parental cells in tibias under docetaxel and/or AMD3100 treatment. $n = 6$ tumors for the control or single treatment group and $n = 8$ tumors for the combined treatment group. **c** Statistical analyses of TRAP occurrence (oc) at the bone-tumor interface in bone samples bearing PC-3 parental cells treated with docetaxel and/or AMD3100. $n = 6$ tumors per group. **d** H&E staining (upper) and TRAP staining (lower) of tibias at 38 days (DU 145) or 43 days (PC-3) after inoculation of cells expressing $KLF5^{KR}$ (KR) or $KLF5^{KQ}$ (KQ). **e** The ratio of tumor area to total sample area of DU 145 (upper) and PC-3 (lower) cells expressing $KLF5^{KR}$ (KR) or $KLF5^{KQ}$ (KQ) in tibias under the treatments of docetaxel and/or AMD3100. For DU 145, $n = 6$ tumors for the control or single treatment group and $n = 8$ tumors for the combined treatment group. For PC-3, $n = 8$ tumors for each group. **f** Statistical analyses of TRAP occurrence (oc) at the bone-tumor interface in bone samples bearing DU 145 (up) and PC-3 (down) cells expressing different forms of KLF5 treated with docetaxel and/or AMD3100. For DU 145, $n = 6$ tumors per group. For PC-3, $n = 8$ tumors per group. B trabecular bone region, BM bone marrow region, T tumor region. Scale bar, 50 μm. Treatments with docetaxel (10 mg/kg twice a week via i.p.) and/or AMD3100 (3.5 mg/kg/day via i.p.) started at day 11 after tumor inoculation. Representative images shown in **a** and **d** are statistically analyzed in **b**, **c** and **e**, **f**, respectively. In panels **b**, **c**, **e** and **f**, data are shown in mean ± S.E.M. NS not significant; *$p < 0.05$; **$p < 0.01$; ***$p < 0.001$ (two-tailed Student's $t$ test). Source data are provided as a Source Data file.

are osteolytic factors (e.g., PTHrP, IL-11, VEGF, and MMPs) that cause osteolytic resorption[15–19]. Osteolytic resorption in turn releases more TGF-β, and thus forms a vicious cycle with TGF-β to drive bone metastatic lesion. In bone metastasis, acetylation of KLF5 is thus an adaptive PTM event for cancer cells to grow and survive in the TGF-β-enriched bone microenvironment; and Ac-KLF5 thus functions as an effector of TGF-β signaling in organizing downstream molecules to promote osteoclastic bone resorption and expand the living space for tumor cells, adding an essential step in the vicious cycle, i.e., TGF-β/Ac-KLF5/CXCR4/IL-11.

Establishment of the role of the TGF-β/Ac-KLF5 axis in bone metastatic lesions in this study provides a unique opportunity to better understand the mechanisms underlying bone metastasis. For example, while EMT is a cellular process that enables tumor cells to adapt to the bone microenvironment[70], EMT features have been detected in PCa bone metastases[22], and TGF-β is a potent inducer of both EMT and bone metastasis[69,74,75], it remains to be determined whether EMT is essential for bone metastasis of PCa. The in vitro and in vivo systems established in this study could prove useful tools for addressing this important question.

Ac-KLF5-mediated bone metastasis is uncoupled from more rapid cell proliferation. We noticed that absence of KLF5 in PCa cells did not significantly change their propagation in vitro but dramatically attenuated their tumor formation (Figs. 2a–d and 3e–h) by affecting both apoptosis and cell proliferation (Figs. 2g, h and 3i–k), which indicates that KLF5 is essential for tumor formation and is consistent with reported functions of KLF5 in stem cell maintenance and embryonic development[49,76–78]. Interestingly, while acetylation of KLF5 restored the tumorigenicity of PCa cells, it slowed tumor growth[41] (Fig. 3).

It has been long known that TGF-β has dual roles in tumorigenesis, being a tumor suppressor in early stage tumorigenesis by inducing apoptosis under certain conditions but inducing chemoresistance and bone metastasis during tumor progression[25]. What determines the dual functions of TGF-β, or how its function switches between a tumor suppressor and a tumor promoter, is an important question that remains unanswered. A recent study demonstrated that knockdown of *KLF5* enabled TGF-β to induce lethal EMT, i.e., an EMT process followed by apoptosis, and such EMT with repressed KLF5 was considered responsible for TGF-β's tumor suppressor activity in pancreatic carcinoma[79]. When not lethal, however, EMT is closely associated with cancer stemness[35], chemoresistance, and metastases[25,71]. Taken together with our observations that KLF5 was indispensable for tumor formation (Figs. 2a–d and 3e–h), absence or knockdown of *KLF5*

induces EMT (Figs. 2e and 3l, and Supplementary Figs. 2–4), downregulation of KLF5 is indispensable for TGF-β to induce EMT[46], and TGF-β and Ac-KLF5 form a signaling axis to repress cell proliferation while inducing and maintaining EMT, we conclude that, at least in some cancers, the status of KLF5 loss determines whether TGF-β is a tumor suppressor or a tumor promoter. When KLF5 is absent or severely downregulated, TGF-β acts as a tumor suppressor by inducing lethal EMT. Otherwise, TGF-β induces KLF5 acetylation, and the TGF-β/Ac-KLF5 axis induces and maintains EMT and tumorigenicity, prevents apoptosis even under stresses, and causes bone metastasis and chemoresistance. Based on this conclusion, targeting Ac-KLF5 should improve the specificity of therapies using TGF-β inhibitors, because TGF-β is widely distributed in various tissues, and lack of specificity has been a major challenge in targeting TGF-β signaling for the treatment of advanced cancers[36].

The findings in this study could have important implications for the detection and treatment of metastatic PCa. For example, the majority of bone metastases from PCa patients were significantly enriched for cancer cells with increased Ac-KLF5 expression (Fig. 9a, b), which, together with the causal roles of Ac-KLF5 in bone metastasis and resistance to docetaxel (Figs. 2, 8), suggests that detection of Ac-KLF5 in PCa cells, including circulating tumor cells, could help predict the occurrence of bone metastasis and chemoresistance. In addition, taxanes docetaxel and cabazitaxel are the only chemotherapeutic agents that have a survival benefit for patients with mCRPC, but virtually all such patients eventually develop drug resistance[8], and patients with docetaxel-treated bone metastases still have poor prognoses[9]. It is thus urgent to develop therapeutic strategies for treating chemoresistant bone metastases. Considering that Ac-KLF5 directly activates the transcription of *CXCR4* (Fig. 6), the ligand for CXCR4 (i.e., CXCL12) is enriched in the bone environment, CXCR4 promotes bone metastasis and docetaxel resistance[58,80], and CXCR4 was indeed indispensable for Ac-KLF5 to promote bone metastatic lesions and docetaxel resistance (Fig. 8), combined treatments with docetaxel and CXCR4 inhibitors could prove beneficial to patients with PCa bone metastasis. In our tibial model of bone metastasis with Ac-KLF5 expression, inhibition of CXCR4 with plerixafor (i.e., AMD3100), an FDA-approved drug for non-Hodgkin lymphoma and multiple myeloma, not only suppressed bone metastatic growth but also sensitized bone metastases to the therapeutic effect of docetaxel (Fig. 8). Consistently, application of plerixafor also sensitized subcutaneous tumors to docetaxel (Supplementary Fig. 9)[58], but this effect was only detectable in tumors with Ac-KLF5 (Supplementary Fig. 9f–i). Based on these findings, clinical

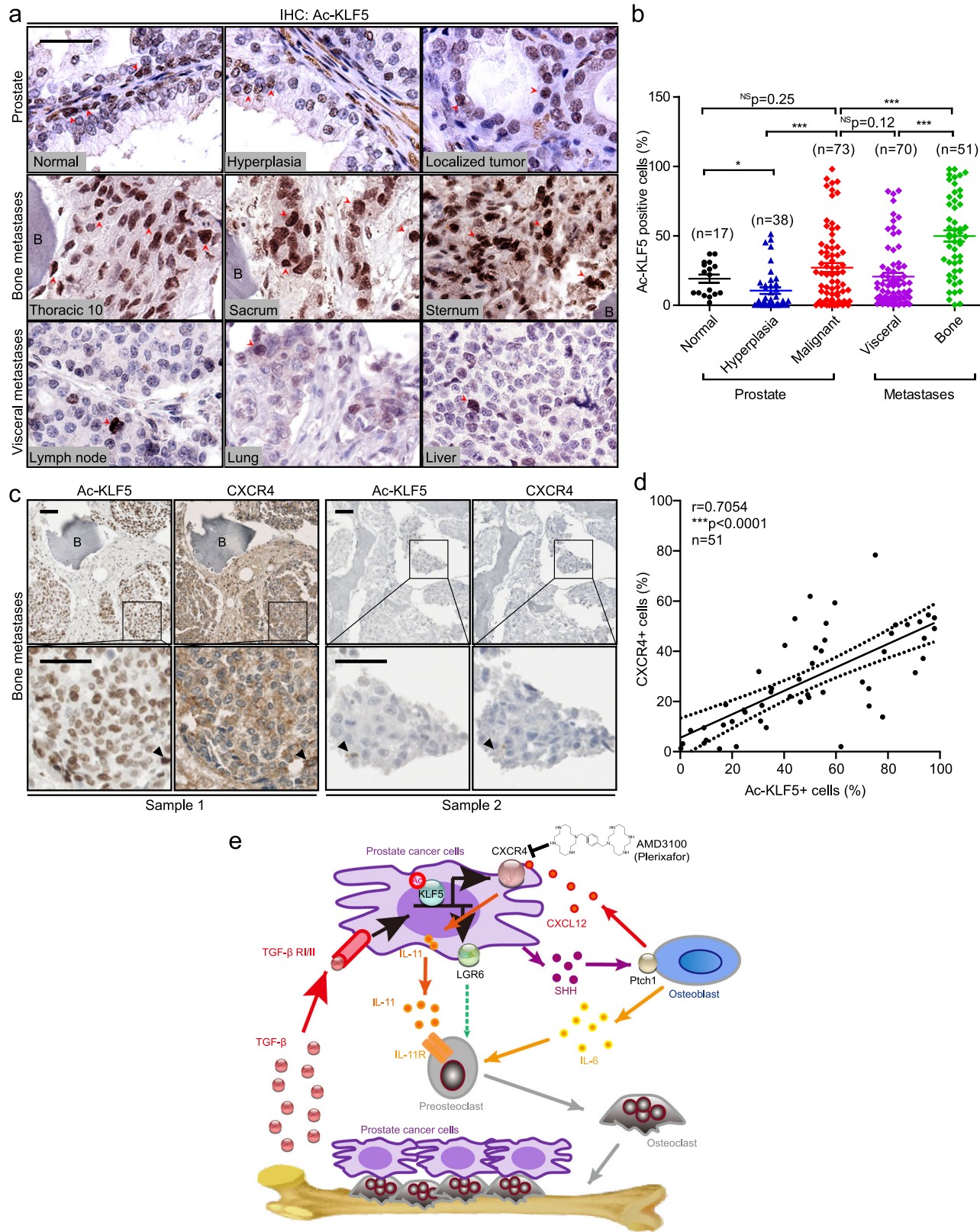

trials are warranted to test whether combined docetaxel and plerixafor treatment provides survival benefits to patients with Ac-KLF5-expressing bone metastases, and detection of Ac-KLF5 would be beneficial for optimizing therapies. Finally, PCa cells expressing different forms of KLF5 and their subcutaneous and tibial xenograft models established in this study could prove useful for the development of therapies targeting bone metastasis.

## Methods

**Cell lines and mouse strains**. Two PCa cell lines, PC-3 and DU 145, were purchased from the American Type Culture Collection (ATCC, Manassas, VA) and propagated according to manufacturer instructions[41]. The C4-2B cell line was a gift of Dr. Leland Chung (Cedars-Sinai Medical Center, Los Angeles, CA). Male BALB/c nude mice and NOD SCID mice (3–4 weeks old) were purchased from Charles River (San Diego, CA) and the Jackson Lab (Bar Harbor, ME) respectively, and closely monitored and handled at an Emory University Division of Animal

**Fig. 9 Acetylated KLF5 is upregulated and positively correlated with CXCR4 in bone metastases of PCa. a** Representative images of IHC staining of Ac-KLF5. Red arrows indicate positive Ac-KLF5 staining. Scale bar, 50 μm. **b** Quantitative analyses of Ac-KLF5 expression in benign tissues, hyperplasia, and localized tumors of the prostate and metastases of PCa from both the visceral and bone tissues. Numbers of samples (*n*) are indicated in the figure. Data are shown in mean ± S.E.M. NS not significant; *$p < 0.05$; ***$p < 0.001$ (two-tailed Student's *t* test). **c** Ac-KLF5 expression is positively correlated with CXCR4 expression in bone metastases of PCa patients, as indicated by representative images of IHC staining in two bone metastasis samples. Black arrow heads indicate stronger Ac-KLF5 and CXCR4 staining in the same cells from consecutive sections. Scale bar, 50 μm. **d** Pearson analyses of the expression of Ac-KLF5 and CXCR4 in 51 bone metastasis samples of PCa patients. ***$p < 0.0001$. **e** A schematic depicting how acetylation of KLF5 leads to bone metastasis by promoting osteoclast differentiation via transcriptionally activating CXCR4, which in turn increases IL-11 secretion. Representative images shown in **a** and **c** are statistically analyzed in **b** and **d**, respectively. Source data are provided as a Source Data file.

---

Resources facility according to the policies of the Institutional Animal Care and Use Committee. For the animal housing condition, the default specification for temperature is controllable within a range of 65-86 °F and ±1 °F of the set point year-round and the relative humidity is controlled within a range of 40–50% and within 10% of the set point year-round. By default, animal housing areas are on a 12 h × 12 h light/dark cycle. The animal studies were approved by the Institutional Animal Care and Use Committee (IACUC) of Emory University.

**Knockout of *KLF5* in PCa cell lines**. The CRISPR-cas9 system was used to eliminate KLF5 protein according to the protocol from the Feng Zhang laboratory[81]. Briefly, sgRNA-encoding DNA was designed and synthesized as DNA oligos specific for the KLF5 genome: 5′-CACCGACGGTCTCTGGGATTTGTAG-3′ and 5′-AAACCTACAAATCCCAGAGACCGTC-3′, annealed and then cloned into the CRISPR-cas9 lentivirus backbone lentiCRISPRv1 vector (Addgene, Boston, MA, #49535), and lentiviruses were generated following the protocols described on the Addgene website (http://www.addgene.org/lentiviral/protocols-resources/). PCa cells infected with lentiviruses were selected in the medium containing puromycin (1 μg/ml) for 72–96 h before seeding and screening for single clones. KLF5-null clones were identified by Western blotting and confirmed by sequencing the targeted genome region after PCR amplification with primers 5′-CACAATCGA-CAAAATAAGCCTG-3′ and 5′-CAGTAGCTGGTACAGGTGGCCC-3′.

**Retroviral expression of KLF5, KLF5^K369R, and KLF5^K369Q**. The coding regions of wild-type *KLF5* and the acetylation-deficient *KLF5^K369R* mutant were amplified by PCR from the plasmids generated in a previous study[41] with primers 5′-CCAAGCTTATGGACTACAAGGACGACGATGACAAGATGGCTACAAG GGTGCTGAG-3′ and 5′-CCATCGATTCAGTTCTGGTGCCTCTTCAT-3′, except that a FLAG-tag was added to the N-terminus of KLF5 or KLF5^KR. PCR products were digested with HindIII and ClaI restriction enzymes, purified, and subsequently cloned into the pLHCX vector (Clontech, MountainView, CA). The *KLF5^K369Q* mutant, which encodes the Ac-KLF5-mimicking KLF5^KQ, was generated by site-directed mutagenesis with primers 5′-AACCCCGATTTGGAGCAA CGACGCATCCACTA-3′ and 5′-TAGTGGATGCGTCGTTGCTCCAAATCGG GGTT-3′ following standard procedures. In addition, the sgRNA targeting site was also mutated with primers 5′-TCACTCACCTGAGAACTGGGCTGTATAAA TCCCAGAGACCGTG-3′ and 5′-CACGGTCTCTGGGATTTATACAGCCCAG TTCTCAGGTGAGTGA-3′ to introduce nonsense mutations, which helped to avoid cas9-mediated interference. Retroviruses were packaged and applied to infect PCa cells. All plasmids were sequenced to confirm the expected mutations. Prostate cancer cells with different forms of KLF5, including acetylation deficient mutant KLF5^K369R and acetylation mimicking mutant KLF5^K369Q are available upon reasonable request and standard MTA procedures.

**Tibial tumorigenesis assay**. Nude mice or NOD SCID mice were anesthetized with 3% isoflurane and maintained by 2.5% isoflurane, and no toe reflex of muscle tone was present at this point. Both legs were cleaned with 10% povidone/iodine swab/solution, followed by ethanol, repeating three times. Lateral malleolus, medial malleolus, and lower half of tibia with forefinger and thumb was gently grasped, and then leg combination of flexion and lateral rotation was bent, such that the knee was visible and accessible. While firmly grasping the ankle/leg of mouse, 27 g ½ needle was inserted under the patella, through the middle of the patellar ligament, and into the anterior intercondylar area in the top of the tibia. When inserting the needle into the tibia, the syringe was carefully guided through the growth plate using steady and firm pressure with slight drilling action. Upon penetration of the tibial growth plate, the needle was encountered markedly less resistance. We also used a gentle, lateral movement of the needle to ensure the needle was in the tibia and through the growth plate. The movement was limited because the needle was in the proper place within tibia. A volume of 20 μl of cell solution Little (1 million cells for PC-3 and DU 145 cells and 0.3 million for C4-2B cells) was injected into tibia, and no resistance was felt at this point. The needle was then extracted slowly. The mice were placed on a heating pad during the recovery period and monitored every day until the fifth day after injection.

**Histological procedure for bone and TRAP staining**. The mouse bones were fixed in 10% neutral-buffered formalin for 24 h and then decalcified with 10% ethylenediaminetetraacetic acid (EDTA) for 10 days. After decalcification, they were processed in a Tissue-Tek VIP processor and embedded in paraffin for sectioning. Lateral sections (5 μm thick) were cut to include the tibia, knee joint, and the distal femur. Hematoxylin & eosin (H&E) staining and the tartrate-resistant acidic phosphatase (TRAP) staining were done on consecutive sections from each tissue block. For TRAP staining, deparaffinized bone sections were incubated first in 0.2 M acetate buffer for 20 min and then in the same buffer with naphtol AS-MX phosphate (0.5 mg/ml) and fast red TR salt (1.1 mg/ml) for 30-45 min in an 37 °C oven. Color change was monitored every 15 min (TRAP-positive area turns red). Slides were then counterstained with hematoxylin and mounted for analysis with Immu-Mount (Thermo Scientific, Waltham, MA).

**Immunohistochemistry and immunofluorescence staining**. Formalin-fixed paraffin-embedded tissues were sectioned at 5 μm, deparaffinized in xylene, rehydrated in graded ethanol, subjected to antigen retrieval by boiling the slides in a pressure cooker for 3 min in a citrate buffer (10 mM trisodium citrate, pH 6.0), treated with 3% $H_2O_2$, permeabilized with 0.5% (vol/vol) Triton X-100, and incubated with 10% goat serum and then with primary antibodies overnight at 4°C. The antibodies included CXCR4 (1:50, Abcam, Cambridge, MA, #ab124824 and 1:100, Millipore, #AB1846), E-cadherin (1:200, Cell Signaling, Danvers, MA, #3195 s), vimentin (1:200, Cell Signaling, #5741 s), Ki67 (1:300, Thermo Fisher, #RM9106s), and c-caspase 3 (1:250, Cell Signaling, #9661 s). Tissue sections were then incubated with EnVision Polymer-HRP secondary antibodies (Dako, Glostrup, Denmark) at room temperature for 1 h. After the application of DAB-chromogen, tissue sections were stained with hematoxylin, dehydrated, and mounted. For staining of Ac-KLF5 and total KLF5, the antibodies have been generated in our previous studies[39,82,83] and were used at a dilution of 1:250 and 1:4000 respectively. For the staining of Ac-KLF5, antigen unmasking was done in a TE buffer (10 mM Tris and 1 mM EDTA, pH 9.0). For the staining of total KLF5, antigen unmasking was also done in a citrate buffer. Staining intensities of proteins were quantified by using the Fiji software[84].

For immunofluorescence staining of E-cadherin and vimentin, cells were fixed in 4% paraformaldehyde for 30 min, permeabilized with 0.5% (vol/vol) Triton X-100 for 10 min, blocked with 2% BSA for 2 h at room temperature, and incubated with a specific antibody (E-cadherin, 1:200; vimentin, 1:200) overnight at 4 °C. After washing, cells were incubated with the secondary antibody conjugated with FITC (anti-rabbit, 1:1000 dilution) for 1 h at 37 °C. DAPI was used to stain nuclei. Fluorescent images were taken with the Leica SP8 confocal microscope at the Integrated Cellular Imaging Core Facility at Winship Cancer Institute.

**RNA-Seq and ChIP-Seq analyses**. For RNA-Seq, DU 145 cells expressing different forms of KLF5 were collected for RNA isolation and proceeded to library construction using an SE50 protocol, and the libraries were sequenced using single-end 50 bp reads on a BGISEQ-500 at BGI (Shenzhen, Guangzhou, China). PC-3 with different forms of KLF5 were collected for RNA isolation at the Emory Integrated Genomics Core, and proceeded to library construction and RNA-Seq at Novogene (Sacramento, CA) using paired-end 150 bp reads on a NovaSeq. FASTQ files from sequencing were quality controlled and adapter trimmed using FASTQC (v0.11.5) and mapped to HG19 reference genome using the STAR aligner (v2.5.0a)[85]. Putative PCR duplicates were marked and removed with SAMtools (v1.7) for downstream analysis[86]. Gene expression levels were determined by the number of fragments per kilobase per million reads (FPKM), similar to a previously described procedure[87]. Briefly, reads overlapping exonic regions of UCSC HG19 known genes were determined using the 'summarizeOverlaps' function of the 'GenomicAlignments'_(v1.20.1) package in the R/Bioconductor (v3.6.1). Differentially expressed genes were determined using edgeR (v3.26.5) with an FDR ≤ 0.05 determining significance[88].

DU 145 cells with different forms of KLF5 (40 million cells for each) were used for ChIP assay following the procedures of SimpleChIP® Enzymatic Chromatin IP Kit (Magnetic Beads) (Cell Signaling, #9003). The CelLytic NuCLEAR Extraction Kit (Sigma, St. Louis, MO, # NXTRACT) was used for nuclear extraction to enhance ChIP efficiency. Nuclear fraction of cells was digested by micrococcal nuclease (provided in the kit of ChIP assay) at 37 °C for 20 min. The KLF5

antibody from R&D (Minneapolis, MN, #AF3758) (60 μg) was used to pull down KLF5 bound sequences. The DNA samples were sent to BGI for quality control, library construction (SE50), and sequencing with the BGISEQ-500 sequencer. FASTQ files of sequencing were quality controlled and adapter trimmed using FASTQC (v0.11.5) prior to mapping them to the HG19 reference genome using bowtie (v2.2.6)[89]. Enriched regions were determined using MACS2 (v2.1.0.20151222) relative to input files with a *q* value (FDR adjusted *p* value) of 0.05[90]. The union of all enriched regions was determined and coverage in these regions were determined using the 'summarizeOverlaps' function in R/Bioconductor (v3.6.1) prior to differential analysis using edgeR (v3.26.5) where regions with an FDR ≤ 0.05 were considered significant[88].

The RNA-Seq and ChIP-Seq data have been uploaded into Gene Expression Omnibus with an accession number of GSE161951, which is linked to two SubSeries, GSE161949 for ChIP-Seq and GSE161950 for RNA-Seq.

**Real-time qPCR and ChIP-qPCR**. Total RNA was isolated from cells using the Trizol reagent (Invitrogen, Carlsbad, CA), and the first strand cDNA for mRNA was synthesized from total RNA using the RT-PCR kit from Promega (Madison, WI). Real-time qPCR was performed to detect the expression of markers for EMT and osteoclast differentiation and KLF5-regulated paracrine factors. Primers for these analyses are listed in Supplementary Table 1. ChIP with the KLF5 antibody from R&D (#AF3758) was performed using the SimpleChIP® Enzymatic Chromatin IP Kit (Cell Signaling, #9003) following the manufacture's instructions. Primers for these analyses are listed in Supplementary Table 3. The SYBR green (Takara, Mountain View, CA) method was used with the 7500 Fast Real-Time PCR System (Applied Biosystems) to detect gene expression. Real-time qPCR reactions were performed in duplicate or triplicate.

**Western blotting**. Western blotting was performed following general protocol provided in the website of Cell Signaling Technology (https://www.cellsignal.com/contents/resources-protocols/western-blotting-protocol/western). The KLF5 (1:1000 in 5% nonfat dry milk) and Ac-KLF5 (1:50 in 2.5 % BSA) antibodies were generated in our previous studies[39,82,83]. The antibodies for E-cadherin (#3195, 1:1000) and vimentin (#5741, 1:1000) were purchased from Cell Signaling, that for N-cadherin (#610920, 1:1000) was from BD Biosciences (Franklin Lakes, NJ), and that for β-actin (A2066, 1:5000) was from Sigma.

**Flow cytometry**. The expression of CXCR4 in human PCa cells with different forms of KLF5 was also measured by flow cytometry. Single cell suspensions were stained with PE-conjugated anti-CXCR4 antibody (Biolegend, San Diego, CA, #306506) in dark overnight at 4 °C. PE-conjugated anti-mouse IgG2α, κ Isotype (Biolegend, #400211) was used as the control antibody. Data was plotting using Flowjo software (v7.6.1).

**Luciferase reporter assay**. Truncated promoters of *CXCR4* were cloned into the pGL3 basic plasmid (Promega) by using primers listed in Supplementary table 2. All plasmids were sequenced to confirm truncations of sequences. The *CXCR4* promoter activity assay was carried out in 293T cells using 0.2 μg plasmids. Cells were cotransfected with 0.4 μg of pLHCX-KLF5 plasmids (or its mutants) and 0.005 μg pGL4.70 (Renilla luciferase, Promega). Renilla luciferase served as an internal control. The Lipo2000 reagent (Invitrogen) was used for plasmid transfection according to the manufacturer's instructions. Forty-eight hours after transfection, cells were lysed with 100 μl of Passive Lysis Buffer (Promega), and luciferase activities were measured from 20 μl of cell lysates by using the dual luciferase reporter assay on a Berthold FB12 Luminometer (Berthold, Bad Wildbad, Germany). Luciferase activities were normalized by the renilla luciferase activities. Each data point was in triplicate.

**Migration and invasion assays**. For the migration assay, DU 145 or PC-3 cells were serum-starved overnight and then seeded at 30,000 cells/well onto the upper chamber of a transwell (8 μm, Millipore, Burlington, MA), with medium containing 10% FBS added into the lower chamber. After 24 h at 37 °C in a humidified chamber supplemented with 5% CO₂ in air, transwell membranes were fixed in 4% paraformaldehyde for 1 h, cells on the upper surface of the membrane were scraped by a cotton swab, and cells on the lower surface were stained with 0.1% crystal violet (Sigma) for 0.5 h. Cells were then eluted for 10 min in 250 μl of 10% acetic acid, and the absorbance was measured at 562 nM and divided by the absorbance of an equal number of seeded cells to indicate the migration rates. In cases where only a few cells migrated, photos were taken, the number of migrated cells was counted, and the rate of migration was then calculated based on the number of seeded cells. Each treatment was in duplicate, and each experiment was repeated at least twice. Similar procedures were applied for the invasion assay, in which 50 μl of 1:5 diluted Matrigel (BD) was used to coat the membrane in the upper chamber.

**Sphere formation assay**. For the sphere formation assay of cultured cells, cells were plated at 3000 cells/well into 8-well chamber slides (BD, #354118) precoated with 50 μl Matrigel in serum-free prostate epithelial basal medium (PrEBM) supplemented with 4 μg/ml insulin, B27 (1:50) (Invitrogen), and EGF and bFGF

(each at 20 ng/ml). Images of spheres were taken under microscopy for morphologies after 1–2 weeks of culture, and the total number of spheres as well as the number of spheres with different morphologies (round vs. protruding) were counted. In some experiments, the area occupied by spheres was determined by using the Image J software. For propagation of spheres, spheres in Matrigel were dissolved by dispase (5 mg/ml) (Stemcell, Cambridge, MA, #07913) at 37 °C for half an hour, collected by centrifugation, and digested to single cells with accutase (Stemcell) for 10 min. Single cells were then counted and re-seeded into Matrigel.

To analyze the sphere formation capability for cancer cells from subcutaneous tumor xenografts, cancer cells were firstly isolated into single cells. Briefly, fresh tumor tissues were cut into small pieces in cold medium, digested in Accumax solution (Stemcell) for 3 h at room temperature, and filtered with 40 μm nylon mesh. The cell suspension was then gently loaded onto a layer of Histopaque-1077 gradient (Sigma) and centrifuged at 400×*g* for 30 min at room temperature. Live tumor cells were then collected at the interface and used for sphere formation assay.

**Subcutaneous tumorigenesis assay**. Cancer cells were resuspended in a mixture of PBS and Matrigel (equal volumes) at $2 \times 10^7$ cells/ml for PC-3 and DU 145, and 100 μl of cells were then injected subcutaneously into both flanks of mice. Tumor volumes were measured every 3 (DU 145) or 8 (PC-3) days. After mice were euthanized, tumors were surgically isolated, weighed, photographed, fixed, and sent to the Research Pathology Core Laboratory at Emory for histological analysis.

**Osteoclastogenesis assay**. RAW264.7 preosteoclasts (#TIB-71) were purchased from ATCC, and their differentiation to mature osteoclasts was validated firstly by seeding 10,000 cells in 24-well plate, treating the cells with Rankl (Peprotech, Rocky Hill, NJ) at 20 ng/ml for 6 days with media replenished on day 4, and staining TRAP using a leukocyte acid phosphatase kit (#387 A, Sigma-Aldrich) following the manufacturer's instructions. A limited concentration of Rankl provides a base level osteoclast differentiation.

To test the role of KLF5 acetylation in osteoclast differentiation, RAW264.7 cells were seeded at a density of 10,000 cells in 24-well plates with 5000 PCa cells (DU 145 or PC-3) with different forms of KLF5, and cultured for 6 days in the presence of Rankl at 20 ng/ml. The media were replenished on day 4. TRAP staining was then stained.

To test paracrine signaling from PCa cells, conditioned media (CM) were collected from sub-confluent tumor cells grown in DMEM with 5% FBS for 24 h, and then mixed 1:1 with fresh complete media. RAW264.7 cells were seeded at onto 24-well plates at 10,000 cells per well, cultured for 24 h, and then treated with the CM mixture for 6 days in the presence of Rankl at 20 ng/ml. Media were replenished on day 4, and TRAP was stained as described above.

**Tissue microarrays of normal prostate tissues, PCa, and bone and visceral metastases**. Four tissue microarrays composed of 51 bone metastases and 70 visceral metastases of PCa patients were obtained from the Prostate Cancer Biorepository Network (PCBN) supported by the Department of Defense Prostate Cancer Research Program, and provided by the University of Washington. Three tissue microarrays (PR807c, PR808, and PR8011b) containing 29 normal tissues, 46 hyperplasia tissues, and 117 localized prostate cancers were purchased from US Biomax (Derwood, MD). The PCBN and US biomax have complied with all relevant ethical regulation and been approved by the ethics committees. They have obtained informed consent from all participants. When 2 or 3 cores were available for the same specimen on a microarray, only one of them was used for analysis. Any torn tissue cores were excluded from statistical analysis.

**Statistical analysis**. Readings in all experiments were expressed as means ± standard errors. The statistical significance of differences between two groups was determined by using unpaired Student *t* test, and *p*-values of 0.05 or smaller was considered statistically significant. Two-way Anova tests were used for the analysis of tumor volume curves. Graphpad Prism (v8.0.1) was used for plotting the data and performing statistical analysis.

**Reporting summary**. Further information on research design is available in the Nature Research Reporting Summary linked to this article.

## Data availability

The data generated or analyzed during the current study are available within the article, supplementary information, and attached source data file or from the corresponding author upon reasonable request. The RNA-Seq and ChIP-Seq data that support the findings of this study have been deposited in Gene Expression Omnibus (GEO) with an accession code of G1, which is linked to two SubSeries, GSE161949 for ChIP-Seq data and GSE161950 for RNA-Seq data. Source data are provided with this paper.

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

## Acknowledgements

We thank Dr. Anthea Hammond of Emory University for editing the manuscript. We thank Dr. Dezhi Wang at the Pathology Core Research Lab in the Department of Pathology of University of Alabama at Birmingham for processing bone tissues. We thank Ms. Lingwei Xiang from ICF for statistical input in data analysis. This work was supported by grant W81XWH-18-1-0526 from Department of Defense Prostate Cancer Research Program, grant R01CA171189 from the National Institutes of Health, and grant 81130044 from the National Natural Science Foundation of China. The research reported in this publication was supported in part by the Integrated Cellular Imaging Core Facility, the Research Pathology Core Laboratory, and the Emory Integrated Genomics Core of Emory University Winship Cancer Institute under NIH/NCI award number P30CA138292. The tissue microarray of bone metastasis samples from PCa patients in this work is from the Prostate Cancer Biorepository Network (PCBN) supported by the Department of Defense Prostate Cancer Research Program Awards W81XWH-18-2-0013, W81XWH-18-2-0015, W81XWH-18-2-0016, W81XWH-18-2-0017, W81XWH-18-2-0018, and W81XWH-18-2-0019.

## Author contributions

B.Z. designed and performed most experiments, analyzed the data, and wrote the manuscript; Q.W. and C.F. performed RNA-Seq and ChIP-Seq experiments; Y.L., B.B., L.B., P.M.V., H.R.C., and J.K. performed bioinformatic analyses; L.X., X.L., D.W., W-P.Q., L.Y., and S.X. performed some of the animal experiments; J.C., O.K., and W.Z. helped with some experimental designs; A.O.O. conducted pathological analyses; Y.Z. and M.L. performed some experiments; and J.D. conceived the project, designed and supervised the study, analyzed the data, provided overall guidance, and revised and finalized the manuscript.

## Competing interests

The authors declare no competing interests.
