## [Peer Review File · Nature Communications]

Reviewers' comments:

Reviewer #1 (Remarks to the Author):

The authors previously reported that KLF5 is acetylated in response to TGF-beta stimulation and acetylated KLF-5 has distinct transcriptional activity from that of non-acetylated KLF-5. In this study, the authors found that Ac-KLF-5 is abundant in prostate cancer cells that colonize in bone. Then, they tried to elucidate the role of Ac-KLF-5 and its downstream signaling pathways in metastatic colonization of prostate cancer cells in bone. They showed that the KLF5-KQ mutant that mimics Ac-KLF-5 induces activation of a network of cytokine signaling including CXCR4 and IL-11, leading to bone lesions and chemoresistance. The work for the first time addressed in vivo function of Ac-KLF5 and could contribute to the research field on TGF-beta and cancer progression. Generally, experiments were well performed.

I have one serious concern. The main conclusion of this work is based on experiments using cells expressing acetylation-mimicking mutant of KLF5 (KLF5-KQ). The authors successfully unveiled the importance of CXCR4 and IL-11 downstream of KLF5 by use of the mutant. That is OK. However, the authors have previously reported that TGF-beta transcriptionally downregulates KLF5 in other epithelial cell lines (ref 45). In addition, TGF-beta also downregulates KLF5 in prostate cancer cells, DU145 and PC-3 (Figure 1D and E), which were used whole through this study. Because KLF5 mutants were expressed using the CMV promoter, they are resistant to transcriptional down-regulation by TGF-beta. Therefore, I am afraid that most of the data are obtained in very artificial situations. I feel that, at least, some of the key experiments including Figure 7 should be performed using parental DU145 or PC-3 cells.

Other comments are as follows:

1) Figure 1 D: KLF-5 in WCL is strikingly downregulated by TGF-beta stimulation, but KLF5 in immunoprecipitates is almost constant. Thus, the KLF-5 IP data are not quantitative and only show the AcKLF5/KLF5 ratio. Here, detecting total amount of Ac-KLF-5 is important because it should be up-regulated if Figure 1A is true. Therefore, Ac-KLF-5 in WCL should be detected using anti-Ac-KLF-5. Fortunately, the authors have the antibody.

2) The authors examined effects of KLF mutants on EMT marker expression in Figure 1F-G (in vitro) and Figure 2E-F (in vivo). In the former in vitro experiments, they used DU145 cells ectopically expressing KLF5 or its mutants. In the latter in vivo experiments, they used KLF5 knockout DU145 and PC-3 cells, to which KLF5 or its mutants were re-introduced. This is quite confusing because in the former, KLF5 WT and KR mutant behaved similarly while in the latter, WT and KQ mutant did so.

To avoid such confusion, data in Figure 1F-H should be replaced by those using KLF5 knockout DU145 cells.

3) Figure 1 F-H: It remains unclear if KLF5 KQ mutant itself is sufficient to exert downstream events or still requires TGF-beta signaling additionally. Cells not treated with TGF-beta should also be examined.

4) Figure 7: Here the therapeutic effects of docetaxel and/or AMD3100 were examined using cells expressing KR or KQ mutant. However, I feel that the conditions are very artificial as mentioned above. It is important to see the effects using parental DU145 or PC-3 cells.

Minor comments:

5) Figure 1D and E: Why did the authors use anti-AcK, not Ac-KLF-5? The band may be Ac-KLF-5, but also may be acetylated protein bound to KLF-5. Because the authors have an antibody to detect Ac-KLF-5, they should have used it. I would also like to confirm that the experiments were performed in the absence of trichostatin A. Right? The authors previously used trichostatin A to detect Ac-KLF5 in Ref 39.

6) The Discussion section is too long and can be shortened.

Reviewer #2 (Remarks to the Author):

In this paper, Zhang and colleagues proposes that KLF5 (a transcription factor) acetylation promotes prostate cancer (PCa) bone metastasis by activating CXCR4/IL-11 signaling axis. This group has been studying KLF5 in PCa since 2001 and has previously demonstrated that Klf5 deletion promotes the development and growth of murine prostate tumors initiated by Pten deletion (Xing C et al., Neoplasia 16:883, 2014), suggesting that KLF5 behaves as a tumor suppressor in certain context of prostate tumorigenesis. Klf5 acetylation has been implicated in self-renewal and differentiation of mouse ES cells (Zhao et al., PLoS ONE, 10:e0138168, 2015) and androgens have been shown to induce KLF5 leading to increased CXCR4-dependent prostate cancer cell migration (Frigo DE et al., Mol Endocrinol, 23:1385-96, 2009). In this project, the authors show that TGF β , putatively produced in the bone, promotes KLF5 acetylation (Ac-KLF5 or AcK), which in turn promotes osteoclastogenesis and bony metastasis by activating CXCR4/IL-11/SHH/IL-6 signaling. Virtually all studies and results were from studies using PC3 and Du145 cells and the true translational value and clinical relevance

remains unclear. The potential significance of the study is further diminished by some technical and experimental concerns and, frequently, very subtle effects, as exemplified below.

Figure 1 & Fig. S1:

- 1) Images in Fig. 1A are of poor quality and Ac-KLF5 nuclear staining isn't that clear, casting doubt on the quantitative data in Fig. 1B.
- 2) Authors should have stained KLF5 side-by-side for comparisons.
- 3) Furthermore, In Fig. 1D-E, Western blotting using WCL clearly showed decrease/loss of total KLF5 but, mysteriously, the IP of KLF5 showed no changes in total KLF5.
- 4) Some of the blotting data in Fig. 1D-E is inconsistent with that in Fig. S1L.
- 5) In addition, Du145 cells were initially derived from a brain metastasis; therefore, it's unclear how relevant Du145 based studies and results are to human PCa bone metastasis.

Figure 2 & Fig. S2: On the face value, the observations that 'KLF5-null cells had negligible growth in the bone (Fig. 2A, 2C and 2D)' conflicts with their earlier findings that Klf5 functions as a tumor suppressor in the context of Pten deficiency. In Fig. S2G, WB of acKLF5 should be shown.

Figure 4 & Fig. S4: It is unclear how mechanistically Ac-KLF5 induces osteoclastic differentiation.

Figure 5 & Figure 6: A significant amount of studies were carried out in Du145 cells, which were originally derived from a brain metastasis. The relevance of these studies to patient PCa bone metastasis is unclear.

Figure 8: This is very important data. As the authors hypothesize that it's the AcKLF5 that plays a crucial role in the development of chemoresistant PCa metastasis, they show that patient bony metastases demonstrate higher Ac-KLF5/unacetylated KLF5 ratio. Also, correlation between Ac-KLF5 and CXCR4 appears to be subtle, based on the images presented (Fig. 8C).

Reviewer #3 (Remarks to the Author):

This is a very interesting and potentially compelling article elucidating the importance of KLF5 acetylation in response to TGF- β induced tumorigenicity and outgrowth of bone metastasis in prostate cancer. The authors link activation of transcription of CXCR4 and secretion of IL-11 to KLF5 acetylation and promoter activation. While there are many strengths of this manuscript and the work has potential clinical significance, there are some major problems that need to be addressed.

1. Throughout the manuscript there is a failure to demonstrate statistical rigor. Experiments are performed and a sample figure is shown, but there is no reference to how many mice were studied, how many tumors were analyzed, variability between tumors regarding expression of the markers shown. Figure 1 for example does not show how many tumors were analyzed and only one tumor sample appears to have been studied in the Western blots. Moreover it is not clear when 2D cell cultures were analyzed, sphere cultures or tumors are being analyzed. In figures 1I and 1J, the important analysis is the statistical difference between KLF5 and KR and KR to KQ. Greater input from the statistician is needed for analysis of the data throughout the manuscript.

2. In figure 2 it is stated that 5-8 samples were studied for each group, but how many of those were analyzed in E, F, G, and H? How many fields per slide were quantitated and how many slides per tumor were analyzed? What statistical analysis is used?

3. In figure 3 data shown with the KR and KQ mutants illustrate that subcutaneous tumors expressing the KR mutant grow faster for both DU145 and PC3 cells expressing KQ. Given these differences, one would expect that the transcriptional analysis data would come from tumors growing in the tibia--to ensure that the correct microenvironment is reproduced with analyzing the significance of the KR and Q mutants. However, this does not appear to be the case. Figure 4 states that cells and not tibial tumors were analyzed. If indeed these analyses were performed on cultured cells, the significance of the subsequent studies is questionable. Specifically Figure 4 G-J and Figure 5 A-J.

4. In figure 6 A and B --are tibial tumors or cultured cells analyzed for CXCR4 expression? Is the D analysis from cultured cells or tumor? The relevance of the studies in RAW264.7 cells to tibial osteoclasts is questionable. What are the intratibial levels of IL-11 with or without loss of CXCR4 in the PC3 and DU 145 cells? Also it is hard to believe that cells can sense the difference between 510 and 430 pg/ml of IL-11 based upon classic Scatchard analysis of binding to the receptor. While additional IL-11 experiments were performed, it would be more valid to KO the IL-11 receptor and determine effects.

5. In Figure 8 A the stated Ac-KLF5 staining is not apparent at the arrows indicated. It is challenging to distinguish the DAB staining from the nuclear hematoxylin. Increased power is needed.

6. The discussion is way too long.

Point by point responses to reviewers' comments

Manuscript NCOMMS-19-33932-T, by Zhang et al., entitled “Acetylation of KLF5 causes chemoresistant bone metastasis by activating CXCR4/IL-11 while maintaining EMT and tumorigenicity in prostate cancer”

Reviewer #1 (Remarks to the Author):

Comments – overall: The authors previously reported that KLF5 is acetylated in response to TGF-beta stimulation and acetylated KLF-5 has distinct transcriptional activity from that of non-acetylated KLF-5. In this study, the authors found that Ac-KLF-5 is abundant in prostate cancer cells that colonize in bone. Then, they tried to elucidate the role of Ac-KLF-5 and its downstream signaling pathways in metastatic colonization of prostate cancer cells in bone. They showed that the KLF5-KQ mutant that mimics Ac-KLF-5 induces activation of a network of cytokine signaling including CXCR4 and IL-11, leading to bone lesions and chemoresistance. The work for the first time addressed *in vivo* function of Ac-KLF5 and could contribute to the research field on TGF-beta and cancer progression. Generally, experiments were well performed.

Response: We greatly appreciate his/her recognition of the significance of our work in light of addressing the *in vivo* function of KLF5 acetylation in the research field of TGF- β and cancer progression. We also appreciate the reviewer's approval of the quality of our experiments. Our findings indeed establish acetylated KLF5 as an important mechanism for TGF- β -induced bone metastasis in prostate cancer and likely other cancers.

Comment – major concern: I have one serious concern. The main conclusion of this work is based on experiments using cells expressing acetylation-mimicking mutant of KLF5 (KLF5-KQ). The authors successfully unveiled the importance of CXCR4 and IL-11 downstream of KLF5 by use of the mutant. That is OK. However, the authors have previously reported that TGF-beta transcriptionally downregulates KLF5 in other epithelial cell lines (ref 45). In addition, TGF-beta also downregulates KLF5 in prostate cancer cells, DU145 and PC-3 (Figure 1D and E), which were used whole through this study. Because KLF5 mutants were expressed using the CMV promoter, they are resistant to transcriptional down-regulation by TGF-beta. Therefore, I am afraid that most of the data are obtained in very artificial situations. I feel that, at least, some of the key experiments including Figure 7 should be performed using parental DU145 or PC-3 cells.

Response: While the reviewer noticed that total KLF5 was downregulated by TGF- β in both noncancerous epithelial cells (Ref. 45) and prostate cancer cells (New Fig. 1g-i), (s)he did not acknowledge the simultaneous upregulation of acetylated KLF5 (Ac-KLF5) by TGF- β in the same cells (New Fig. 1g-i), yet upregulation of Ac-KLF5 by TGF- β (also occurred in bone metastases of human prostate cancer) is the cornerstone of our study. It is important to point out that downregulation of total KLF5 and upregulation of its acetylated form (Ac-KLF5), while occurring simultaneously, are two distinct processes that indicate two distinct effects of TGF- β on KLF5. In

other words, while downregulation of total KLF5 indicates KLF5 as an epithelial factor, simultaneous induction/emergence of Ac-KLF5 brings out Ac-KLF5 as a totally different player in multiple cellular processes, including EMT, drug resistance, and bone metastasis (New Fig. 2, 8 and Supplementary Fig. 4). To elucidate the biological functions of Ac-KLF5, we have to establish a model to exclude the effects of TGF- β on KLF5 downregulation.

Under this context, the best approach, as we have taken, would be to delete the endogenous *KLF5* (as we have done using the CRISPR-Cas9 approach) and then ectopically express each KLF5 mutant (Ac-KLF5 mimicking KLF5^{K369Q} or acetylation-deficient KLF5^{K369R}).

Another design that addresses this concern is that we have also used the acetylation-deficient KLF5^{K369R} mutant as a control, along with the wildtype KLF5 and empty vector. Different effects were detected for each form of KLF5, which further validates our approach.

Therefore, although we agree that using parental PC-3 and DU 145 cells has its own merit in elucidating the role of TGF- β in bone metastasis, we believe that our design better addresses the role of KLF5 acetylation in bone metastasis of prostate cancer.

Nevertheless, in addressing this concern, we have used parental PC-3 cells to perform some key experiments similar to those in original Figure 7 according to the reviewer's suggestion. We inoculated parental PC-3 cells to the tibia and then tested whether the CXCR4 inhibitor AMD3100 sensitizes docetaxel treatment in the parental cells (New Fig. 8a-c, shown on *next page*). Subcutaneous tumors of PC-3 parental cells were also treated with docetaxel and/or AMD3100 for comparison with their tibial counterparts (New Supplementary Fig. 9a, b, shown on *next page*). As we described in the Results section (Line 376 on page 14 in the revised manuscript), "In mice bearing tibial tumors of parental PC-3 cells, in which Ac-KLF5 was induced by enriched TGF- β in the bone (Fig. 1a-f), docetaxel alone slightly suppressed tibial tumor growth, as indicated by the reduced tumor size (Fig. 8a, b); and addition of CXCR4 inhibitor AMD3100 (plerixafor) significantly sensitized the tibial tumors to docetaxel, as demonstrated by alleviated bone lesions and reduced tumor area (Fig. 8a, b). The coefficient of drug interaction (CDI) of docetaxel and AMD3100 was 0.54 (less than 0.7), suggesting that combined docetaxel and AMD3100 treatment had a synergistic effect on PCa bone metastasis. As expected, inhibition of CXCR4 by AMD3100 suppressed osteoclast differentiation in tibial tumors (Fig. 8a, c). For subcutaneous tumors, however, AMD3100 failed to sensitize tumors to docetaxel, although docetaxel alone showed a suppressive effect on tumor growth (Supplementary Fig. 9a, b). Subcutaneous tumors expressed less Ac-KLF5 than their tibial counterparts (Fig. 1a-f), which could explain the differences between tibial and subcutaneous tumors."

Both PC-3 and DU 145 cells respond to TGF- β similarly, TGF- β induced the acetylation of KLF5 in both cell lines (New Fig. 1g-i), all functional results with KLF5 mutants are similar between PC-3 and DU 145 (New Fig. 2, 3, 8), and the role of TGF- β in prostate cancer bone metastasis has been well established. Therefore, we only used the PC-3 cell line in the experiments of paper revision.

Figure 8

Supplementary figure 9

Other comments:

Comment 1: Figure 1 D: KLF-5 in WCL is strikingly downregulated by TGF-beta stimulation, but KLF5 in immunoprecipitates is almost constant. Thus, the KLF-5 IP data are not quantitative and only show the AcKLF5/KLF5 ratio. Here, detecting total amount of Ac-KLF-5 is important because it should be up-regulated if Figure 1A is true. Therefore, Ac-KLF-5 in WCL should be detected using anti-Ac-KLF-5. Fortunately, the authors have the antibody.

Response: Sorry for the confusion but the two cannot be compared in this way because of the difference in experimental design. We ran a gel with equal volumes for all samples and detected total KLF5, and the amount of total KLF5 decreased with TGF-β treatments. When performing the IP experiment with KLF5 antibody, we used total KLF5 as the input control. We then increased the amount of IP products for samples with TGF-β treatments to ensure the input amount of total KLF5 similar among samples, and then used antibody against Ac-K to detect Ac-KLF5. The relative levels of KLF5 acetylation in IP products were then calculated accordingly. This adjustment makes the detection of expression differences in Ac-KLF5 among different groups more accurate by minimizing the inaccuracies when quantifying band intensities from films. This is a common practice in detecting modification levels of proteins.

To address and avoid this concern, we have detected the level of Ac-KLF5 at K369 by using the Ac-KLF5 antibody developed in our previous study¹ (New Fig. 1g-i, shown on next page). The original IP assay for the detection of KLF5 acetylation level has been removed.

Figure 1

Supplementary figure 2

Comment 2: The authors examined effects of KLF mutants on EMT marker expression in Figure 1F-G (in vitro) and Figure 2E-F (in vivo). In the former in vitro experiments, they used DU145 cells ectopically expressing KLF5 or its mutants. In the latter in vivo experiments, they used KLF5 knockout DU145 and PC-3 cells, to which KLF5 or its mutants were re-introduced. This is quite confusing because in the former, KLF5 WT and KR mutant behaved similarly while in the latter, WT and KQ mutant did so. To avoid such confusion, data in Figure 1F-H should be replaced by those using KLF5 knockout DU145 cells.

Response: We thank the reviewer for identifying this issue. The inconsistency was caused partially by the variation in total KLF5 expression levels. In the old Figure 1F-1H, ectopically expressing KLF5 or its mutants was performed in parental cells, but in the latter experiments, KLF5 and its mutants were expressed in KLF5-null cells. Therefore, the reviewer is correct. According to the suggestion, we have detected EMT features of DU 145 cells with different forms of KLF5 based on parental cells (New Supplementary Fig. 2, shown on *last page*) and KLF5-null cells (New Supplementary Fig. 4, shown *below*) with or without TGF- β treatment. In the condition of TGF- β treatment, parental cells with overexpression of wildtype KLF5 and the KR mutant behaved similarly (New Supplementary Fig. 2, shown on *last page*). However, when TGF- β was added, the EMT features of cells with wildtype KLF5 were more like the KQ cells on the background of KLF5 knockout, as indicated by epithelial marker E-cadherin and mesenchymal marker vimentin (New Supplementary Fig. 4, shown *below*). Therefore, the EMT phenotypes of the same cells are consistent between *in vitro* (New Supplementary Fig. 4) and *in vivo* (New Fig. 2e, f). These data have been included in the supplementary information according to the overall logic of this story.

Comment 3: Figure 1 F-H: It remains unclear if KLF5 KQ mutant itself is sufficient to exert downstream events or still requires TGF-beta signaling additionally. Cells not treated with TGF-beta should also be examined.

Response: We appreciate the comment. In the revised manuscript, we have included the condition without TGF- β treatment according to this suggestion (New Supplementary Fig. 2, shown on *page 4*). As we described in the Results section (Line 152 on *page 8* in the revised manuscript), “the KLF5^{KQ} mutant itself induced EMT, as indicated by lower E-cadherin, higher vimentin and enhanced migration and invasion capabilities (Supplementary Fig. 2a-d). Furthermore, the KLF5^{KQ} mutant maintained but the KLF5^{KR} mutant abolished TGF- β -induced EMT, as indicated by the expression of EMT markers E-cadherin and vimentin, morphological changes, and the migration and invasion capability (Supplementary Fig. 2a-d).”

Comment 4: Figure 7: Here the therapeutic effects of docetaxel and/or AMD3100 were examined using cells expressing KR or KQ mutant. However, I feel that the conditions are very artificial as mentioned above. It is important to see the effects using parental DU145 or PC-3 cells.

Response: As we explained above in the response to the comment – overall, we have tested the therapeutic effects of docetaxel and/or AMD3100 on parental PC-3 cells (New Fig. 8a-c and Supplementary Fig. 9a, b, shown on *page 3*). We agree with the reviewer that this experiment has its own merit, although our previous design better highlights the role of KLF5 acetylation.

Minor comments:

Comment 5: Figure 1D and E: Why did the authors use anti-AcK, not Ac-KLF-5? The band may be Ac-KLF-5, but also may be acetylated protein bound to KLF-5. Because the authors have an antibody to detect Ac-KLF-5, they should have used it. I would also like to confirm that the experiments were performed in the absence of trichostatin A. Right? The authors previously used trichostatin A to detect Ac-KLF5 in Ref 39.

Response: We appreciate the reviewer for these comments. However, we'd like to point out that, similar to many other posttranslational modifications of proteins, the level of Ac-KLF5 is rather low even after TGF- β treatment; and our anti-Ac-KLF5 antibody is much less efficient when used in western blotting (an 1:50 dilution is required for the antibody). Even with an 1:50 dilution of the anti-Ac-KLF5 antibody, its detection by western blotting is still not as efficient as in IHC staining, which is why we had to use trichostatin A in our previous study (Ref. 39). We eventually found that immune-precipitating with total KLF5 antibody and then blotting with anti-AcK antibody is much more efficient for the detection of Ac-KLF5, as presented in old Figure 1D and E.

We also agree with the reviewer's concern and have performed this experiment as suggested by using antibodies targeting Ac-KLF5 to exclude the possibility raised by the reviewer (New Fig. 1g-i, shown on *page 4*). We did not add trichostatin A when performed the experiments, instead, we used a large amount of proteins.

Comment 6: The Discussion section is too long and can be shortened.

Response: We have significantly shortened the Discussion in the revised manuscript.

Reviewer #2 (Remarks to the Author):

Comments – overall: In this paper, Zhang and colleagues proposes that KLF5 (a transcription factor) acetylation promotes prostate cancer (PCa) bone metastasis by activating CXCR4/IL-11 signaling axis. This group has been studying KLF5 in PCa since 2001 and has previously demonstrated that Klf5 deletion promotes the development and growth of murine prostate tumors initiated by Pten deletion (Xing C et al., *Neoplasia* 16:883, 2014), suggesting that KLF5 behaves as a tumor suppressor in certain context of prostate tumorigenesis. Klf5 acetylation has been implicated in self-renewal and differentiation of mouse ES cells (Zhao et al., *PLoS ONE*, 10:e0138168, 2015) and androgens have been shown to induce KLF5 leading to increased CXCR4-dependent prostate cancer cell migration (Frigo DE et al., *Mol Endocrinol*, 23:1385-96, 2009). In this project, the authors show that TGF β , putatively produced in the bone, promotes KLF5 acetylation (Ac-KLF5 or AcK), which in turn promotes osteoclastogenesis and bony metastasis by activating CXCR4/IL-11/SHH/IL-6 signaling. Virtually all studies and results were from studies using PC3 and Du145 cells and the true translational value and clinical relevance remains unclear. The potential significance of the study is further diminished by some technical and experimental concerns and, frequently, very subtle effects, as exemplified below.

Response: We appreciate the reviewer for his/her nice summary of both the background and our story. The major concern raised by the reviewer is that “Virtually all studies and results were from studies using PC3 and Du145 cells and the true translational value and clinical relevance remains unclear”.

As recognized by the reviewer, the PC-3 cell line was derived from a human prostate cancer metastatic to the bone², and this line has been widely used as a cell model for prostate cancer bone metastasis^{3, 4, 5, 6, 7, 8}. For example, in a recent *Cancer Cell* paper where the role of TGF- β in prostate cancer bone metastasis was examined, the PC-3 model was the only one used⁹. It is important to note that, in our study, we have already used the PC-3 model for all key experiments, in addition to the use of DU 145 cells. Some of our PC-3 data were previously in the supplementary data section, and we have placed them in the main figures to eliminate any confusion. In the revised manuscript, almost all the main figures (New Fig. 1, 2, 3, 4, 6, 7, 8) contain PC-3 data.

More importantly, we have demonstrated that data from both the DU 145 and PC-3 models are consistent in all key experiments, which provides another line of evidence for the relevance of our findings to human prostate cancer bone metastasis.

Further supporting the translational value and clinical relevance, we found that Ac-KLF5 was significantly upregulated in bone metastases of prostate cancer patients (New Fig. 9a, b). This line of investigation was apparently appreciated by all reviewers, as it provides first-hand evidence for the clinical relevance of our study.

Furthermore, CXCR4, a known player in TGF- β -mediated bone metastasis, was upregulated by the Ac-KLF5 mimicking KLF5^{K369Q} mutant but not by the acetylation-deficient KLF5^{K369R} mutant (New Fig. 6); and treatment of docetaxel-resistant bone metastases in a preclinic tibia injection animal model with plerixafor, an FDA-approved CXCR4 inhibitor (also known as AMD3100), clearly sensitized the metastases to docetaxel (New Fig. 8). These findings further validate the translational value and clinical relevance of our study. The therapeutic data

in the New Figure 8 will lead to a clinical trial of treating patients with docetaxel resistant bone metastasis with the combination of docetaxel and plerixafor (AMD3100), because such patients are rather common in the clinic. Therefore, the translational value and clinical relevance are apparent and well supported.

As we know, it'd be ideal if patient-derived xenograft (PDX) models could be developed and used for the study of prostate cancer bone metastasis. However, this is currently unlikely because once bone metastases develop in a patient, surgery is no longer an option of treatment, and no fresh tissues could be isolated for the establishment of PDX models. The bone metastases on the tissue microarray used in our analysis of Ac-KLF5 expression were from autopsies of prostate cancer patients.

Taken together with the reviewer's specific comment 5 below, this concern appears to have originated from our use of the DU 145 cell line, which was originally derived from a brain metastasis rather than bone metastasis. In this regard, we would like to emphasize that, while the DU 145 cell line was indeed derived from a brain rather than bone metastasis of a prostate cancer patient, there are several lines of investigations that indicate the relevance of DU 145 to bone metastasis. **(1)** Injection of DU 145 cells to the tibia has been used in multiple publications as a model of prostate cancer bone metastasis^{10, 11}, including one study in which targeting TGF- β signaling inhibits bone metastatic lesions¹². **(2)** The same patient whose brain metastasis gave rise to the DU 145 cell line also had bone metastasis¹³, which is perhaps unknown to the reviewer. As seen for many patients who develop metastases at multiple organs, brain and bone metastases could certainly share some characteristics responsible for the tumor's metastases. **(3)** Both PC-3 and DU 145 cell lines respond to TGF- β signaling, which has been documented in a number of studies including our own^{1, 12, 14, 15}, and TGF- β is perhaps the best known single factor that causes bone metastasis. Ac-KLF5 is tested as a mechanism for TGF- β to induce bone metastasis in our study, so the TGF- β responsiveness is an important indicator for the relevance of this cell line to prostate cancer bone metastasis. Therefore, although less frequently used than the PC-3 model, DU 145 cells have properties of bone metastasis.

Nevertheless, in further addressing this concern, we have performed key experiments using the C4-2B prostate cancer cell line (New Fig. 1i, 2a-d and 4a-d, shown on *page 4* and *next page*), which was derived from a bone metastasis of the LNCaP human prostate cancer cells^{16, 17} and more importantly, has been more widely used than the PC-3 line in the study of prostate cancer bone metastasis^{10, 18, 19, 20, 21, 22, 23, 24, 25}. In addition, the C4-2B line has often been used with the PC-3 line^{10, 18, 23} and sometimes also with DU 145¹⁰.

Similar strategies were used to deplete endogenous KLF5 in C4-2B cells and establish C4-2B cells with different forms of KLF5 (New Supplementary Fig. 3b, c, f, g, shown on *page 15*). Our findings from C4-2B cells are consistent with those from PC-3 and DU 145 cells: **(1)** TGF- β induced KLF5 acetylation, while downregulating total KLF5 in C4-2B cells (New Fig. 1i, shown on *page 4*). **(2)** Interruption of KLF5 acetylation by mutation effectively suppressed bone metastatic tumor growth and Ac-KLF5 mimicking mutant KLF5^{KQ} caused significant tibial tumor growth in C4-2B cells, as indicated by bone lesions and tumor areas in the bone (New Fig. 2a-d, shown on *next page*). **(3)** The KLF5^{KQ} mutant also induced more extensive osteoclast differentiation than the KLF5^{KR} mutant in C4-2B cells in both tibial tumors and cocultures with preosteoclast RAW264.7 cells (New Fig. 4a-d, shown on *next page*). Experiments with C4-2B cells further improve the clinical relevance of our findings.

We also appreciate the reviewer to point out some defects of our experiments and have addressed them point-to-point as below.

Figure 2

Figure 4

Comments on Figure 1 & Fig. S1:

Comment 1: Images in Fig. 1A are of poor quality and Ac-KLF5 nuclear staining isn't that clear, casting doubt on the quantitative data in Fig. 1B.

Response: We have improved the data quality by showing images with higher magnification (New Fig. 1a, b, shown *below*).

Comment 2: Authors should have stained KLF5 side-by-side for comparisons.

Response: We have stained KLF5 and included the data in the New Figure 1 (shown *below*).

Comment 3: Furthermore, In Fig. 1D-E, Western blotting using WCL clearly showed decrease/loss of total KLF5 but, mysteriously, the IP of KLF5 showed no changes in total KLF5.

Response: We ran a gel with equal volumes for all samples and detected total KLF5, and the amount of total KLF5 decreased with TGF- β treatments. When performing the immunoprecipitation (IP) using KLF5 antibody, we used total KLF5 as the input control. We then increased the amount of IP products for samples with TGF- β treatments to ensure the input amount of total KLF5 similar among samples, and then used antibody against Ac-K to detect Ac-KLF5. The relative levels of acetylated KLF5 in IP products were then calculated accordingly. This adjustment makes the detection of expression differences in Ac-KLF5 among different groups more accurate by minimizing the inaccuracies caused by quantifying band intensities from films. This is a common practice in the detection of modification levels of proteins.

We have developed an antibody against acetylated K369 of KLF5 in our previous study¹. This antibody is much less efficient in western blotting (an 1:50 dilution is required for a detection). Even with an 1:50 dilution of the anti-Ac-KLF5 antibody, its detection by western blotting is still not as efficient as in IHC staining, which is why we had to use an alternative method (i.e., IP with total KLF5 antibody and then western blotting with anti-AcK antibody). This alternative is much more efficient for the detection of Ac-KLF5 but less specific, causing confusion. The same concern was also raised by reviewer 1.

To address the concern and avoid the issue, we have detected the level of Ac-KLF5 at K369 by using the Ac-KLF5 antibody developed in our previous study¹ (New Fig. 1g-i, shown on *page 4*). The original immunoprecipitation assay to detect KLF5 acetylation level was removed.

Comment 4: Some of the blotting data in Fig. 1D-E is inconsistent with that in Fig. S1L.

Response: We agree with the reviewer on this concern. In the previous version, the two experiments were not performed at the same time, and TGF- β reagents were from different batches. We have repeated these experiments by using the same batch of TGF- β in the revised manuscript (New Fig. 1g, h, shown on *page 4* and Supplementary Fig. 1j, shown *below*).

Comment 5: In addition, Du145 cells were initially derived from a brain metastasis; therefore, it's unclear how relevant Du145 based studies and results are to human PCa bone metastasis.

Response: As we have discussed in our response to the reviewer's main concern above, there are multiple lines of evidence for the relevance of our DU 145 data to human prostate cancer bone metastasis. In addition to reorganize PC-3 data to the main figure, we have performed key experiments with the C4-2B cell line and include the data in the revised manuscript (New Fig. 1i, 2a-d and 4a-d, shown on *page 4&10*). Also discussed in our response to the reviewer's major concern, the C4-2B cell line provides a more relevant model than DU 145 as it was derived from a bone metastasis of the LNCaP prostate cancer cells and has been widely used as a model of prostate cancer bone metastasis. Specific to the old Figure 1, we also demonstrated that TGF- β induced the acetylation of KLF5 while downregulated total KLF5 in C4-2B cells (New Fig. 1i, shown on *page 4*).

Comments on Figure 2 & Fig. S2: On the face value, the observations that 'KLF5-null cells had negligible growth in the bone (Fig. 2A, 2C and 2D)' conflicts with their earlier findings that Klf5 functions as a tumor suppressor in the context of Pten deficiency. In Fig. S2G, WB of acKLF5 should be shown.

Response: How KLF5 possesses seemingly opposing functions in tumorigenesis is an intriguing question that has been studied by our group for some years. It is not until recently that we have gained some critical insights into this question. Several key points need to be clarified to help understand and address this question.

(1) KLF5 is indispensable for the proliferation and survival of normal prostate epithelial cells. Loss of both *Klf5* alleles in mouse prostates, while not causing morphological changes, caused apoptosis²⁶. In our very latest paper in *Nature Communications*²⁷, we have further demonstrated that Klf5 is essential not only for the differentiation of basal progenitor cells but also for the proliferation of their basal and luminal progenies. Also in our latest *Nature Communications* paper²⁷, we found that Klf5 and its acetylation are essential for the differentiation of basal progenitor cells to luminal cells. Basal progenitor-derived luminal cells with deacetylated KLF5 cannot survive castration. For example, deacetylation of Klf5 in basal progenitors of mouse prostates disrupted their normal differentiation, causing excess luminal cells. Such luminal cells die after castration and cannot regenerate after androgen restoration.

(2) Acetylation of KLF5 slows but is still needed for the proliferation of cancer cells: Although Ac-KLF5 slows the proliferation of prostate cancer cells, which has been demonstrated in our current and previous studies²⁸, it in fact does not stop tumor growth²⁸. This has also been noticed in our current paper (New Fig. 3). KLF5 has also been shown to be necessary for the growth of other tumors including that of the breast²⁹, liver³⁰, and ovary³¹. Lack of KLF5 can induce lethal EMT in some cancer cells³².

(3) Reprogramming in castration resistant prostate cancer (CRPC) could activate the oncogenic function of KLF5: PC-3 and DU 145 cells are AR-null PCa cells and C4-2B is AR-positive but already resistant to androgen deprivation therapy. CRPC tumors consist of populations of cells that have adapted to AR inhibition via myriad somatic alterations to the AR gene and extensive reprogramming of AR cistromes³³. This reprogramming may create an environment where the oncogenic function of KLF5 can manifest.

(4) Absence of Klf5 could affect tumor-infiltrating immune cells including macrophages, T cells and B cells. As suggested by our unpublished data (Response Fig. 1), decrease in KLF5 expression in human prostate cancer is associated with an increase in infiltrating immune cells including T cells, which are present in genetically engineered mice but absent in nude mice. Therefore, another possibility for Klf5 knockout to promote tumor growth in genetically engineered mice could be attributed to tumor-infiltrating T cells.

(5) Loss of Pten activates the PI3K/AKT survival and proliferation signaling: This activation can overcome cell death induced by many factors during carcinogenesis, which has been well established in a large number of studies. Such an overcome of apoptosis can certainly occur in Klf5-null cells.

Based on these important facts, here is what we propose for how loss of Klf5 in Pten-null mouse prostates promotes Pten loss-induced tumorigenesis: **First**, while loss of Klf5 causes cell death, simultaneous loss of Pten in the same cells activates the PI3K/Akt survival signaling, which prevents the death of Klf5-null cells and leads to the formation of those tumors observed in our previous study³⁴. **Second**, because Klf5 is essential for the differentiation of basal progenitors to luminal cells, as Loss or deacetylation of Klf5 disrupts normal basal to luminal differentiation²⁷, those Klf5/Pten-null basal progenitors not only proliferate uncontrolledly, they also undergo uncontrolled basal to luminal differentiation, leading to those larger and lumen-less tumors as we have observed³⁴.

While remaining to be tested, this hypothesis is supported by several observations: (1) tumors with the deletions of both *Klf5* and *Pten* are indeed large and without typical lumens³⁴; (2) tumors with Pten loss alone have the typical luminal organization and are positive for both

luminal and basal marker expression; (3) tumors with the deletions of both *Klf5* and *Pten* are mostly positive only for luminal marker expression but negative for basal marker expression, which is likely due to excess luminal differentiation after *Klf5* loss.

With regard to the second comment, we agree with the reviewer, and have detected the Ac-KLF5 by Western blotting and include the data in the New Supplementary Fig. 3g of the revised manuscript (shown in *Panel g* of the *below figure*).

Comments on Figure 4 & Fig. S4: It is unclear how mechanistically Ac-KLF5 induces osteoclastic differentiation.

Response: Regrettably the reviewer did not realize that we have clearly deciphered the mechanisms by which Ac-KLF5 induces osteoclast differentiation in old Figures 5 and 6 (New Fig. 5, 6 and 7). In addition, we describe that acetylation of KLF5 activates the transcription of *CXCR4* to promote osteoclastogenesis by increasing IL-11 secretion.

Comments on Figure 5 & Figure 6: A significant amount of studies were carried out in Du145 cells, which were originally derived from a brain metastasis. The relevance of these studies to patient PCa bone metastasis is unclear.

Response: We believe that this concern also caused this reviewer to worry about the relevance and clinical translation of our study, as (s)he commented in the overall summary. Our responses

are the same as those in our responses to his/her overall comment above. Key points include the following: 1) We have also used the PC-3 model, which was derived from a bone metastasis and has been widely used in the study of prostate cancer bone metastasis. 2) Our findings between DU 145 and PC-3 are consistent. 3) DU 145 cells also respond to TGF- β , and our study focuses on the role of Ac-KLF5 in TGF- β -induced bone metastasis. (4) DU 145 cells have been used in multiple publications on prostate cancer bone metastasis^{10, 11, 12}.

In addressing this concern, we have presented almost all PC-3 data in the main figures. Some PC-3 data were in the supplementary figures in the previous version, which might not have been fully appreciated by the reviewer.

In addition, we also performed some key experiments using the C4-2B prostate cancer cell line, which was derived from bone metastasis of the LNCaP prostate cancer cell line (New Fig. 1i, 2a-d and 4a-d, shown on *page 4&10*). The C4-2B data are consistent with those in PC-3 and DU 145 cells, as we described above (shown on *page 9*).

Comments on Figure 8: This is very important data. As the authors hypothesize that it's the AcKLF5 that plays a crucial role in the development of chemoresistant PCa metastasis, they show that patient bony metastases demonstrate higher Ac-KLF5/unacetylated KLF5 ratio. Also, correlation between Ac-KLF5 and CXCR4 appears to be subtle, based on the images presented (Fig. 8C).

Response: We appreciate the reviewer's positive comments on this Figure 8 and recognition of Ac-KLF5 enriched in bone metastases of prostate cancer patients.

We agree that the CXCR4 staining was not as strong with the current antibody. We have used another anti-CXCR4 antibody (Millipore #AB1846), which has been validated for IHC staining already, to repeat the IHC experiments. We also re-analyzed the correlation between Ac-KLF5 and CXCR4 expression after the experiment. The expression of Ac-KLF5 and that of CXCR4 were significantly positively correlated in bone metastases of PCa (New Fig. 9c, d, shown *below*).

Reviewer #3 (Remarks to the Author):

Comments – overall: This is a very interesting and potentially compelling article elucidating the importance of KLF5 acetylation in response to TGF- β induced tumorigenicity and outgrowth of bone metastasis in prostate cancer. The authors link activation of transcription of CXCR4 and secretion of IL-11 to KLF5 acetylation and promoter activation. While there are many strengths of this manuscript and the work has potential clinical significance, there are some major problems that need to be addressed.

Response: We sincerely appreciate the reviewer's recognition of many strengths of our study and overall very positive comments. We have addressed the problems point-by-point as below.

Comment 1: Throughout the manuscript there is a failure to demonstrate statistical rigor. Experiments are performed and a sample figure is shown, but there is no reference to how many mice were studied, how many tumors were analyzed, variability between tumors regarding expression of the markers shown. Figure 1 for example does not show how many tumors were analyzed and only one tumor sample appears to have been studied in the Western blots. Moreover it is not clear when 2D cell cultures were analyzed, sphere cultures or tumors are being analyzed. In figures 1I and 1J, the important analysis is the statistical difference between KLF5 and KR and KR to KQ. Greater input from the statistician is needed for analysis of the data throughout the manuscript.

Response: We appreciate the reviewer pointing out the inadequacy in statistical analyses of our study. In the revised manuscript, we have shown the numbers of mice and tumors in the figure legends. Moreover, we have revised the whole manuscript thoroughly by changing all of the bar charts to plot bar charts to better present the number and the variability of samples. The statistical analyses are also defined at the end of each figure legend. We also included a file titled "Source data" to clarify all the details of the bar charts. We have also consulted with a statistician, Ms. Lingwei Xiang, for validating and double checking our data analysis.

For example, in the legend of the New Figure 1, we now clearly describe that "(c-f) Percentages of Ac-KLF5 (c, e) or KLF5 (d, f) positive cells in subcutaneous and tibial tumors of PC-3 (c, d) or DU 145 (e, f) cells, as counted by Fiji software. For each condition, 3 tumors from 3 mice were used, and 18 different images from the 3 tumors were analyzed, except for Ac-KLF5 in subcutaneous PC-3 and DU 145 tumors and tibial DU 145 tumors, where 12 images were used."

The referenced Western blot data in Figure 1 were from in vitro 2D cell culture, and we apologize for this confusion. In the revised manuscript, we have clarified this issue in the legend of the New Figure 1 and clearly stated that "(g-i) Detection of indicated proteins by Western blotting in whole cell lysates of TGF- β treated PC-3 (g), DU 145 (h) and C4-2B (i) cells in *in vitro* 2-dimensional culture", according to the reviewer's suggestion.

We have also statistically analyzed the differences between KLF5 and KR or between KR and KQ in New Supplementary Figure 2c and d (corresponding to old Fig. 1L and 1J), according to the reviewer's suggestion.

Comment 2: In figure 2 it is stated that 5-8 samples were studied for each group, but how many of those were analyzed in E, F, G, and H? How many fields per slide were quantitated and how many slides per tumor were analyzed? What statistical analysis is used?

Response: We apologize for missing the information. No we clearly describe how many tumors were quantitated in the figure legend of New Figure 2, "(e, f) IHC staining (e) and intensity quantification (f) of epithelial marker E-cadherin and mesenchymal marker vimentin in tibial tumors of DU 145 and PC-3 cells with different forms of KLF5. For DU 145 cells, n=3, 3, 4 and 5 tumors for EV, KLF5, KR and KQ respectively. For PC-3 cells, n= 3, 4, 5 and 5 tumors for EV, KLF5, KR and KQ respectively. (g, h) IHC staining (g) and signal intensity quantification (h) of proliferation marker Ki67 in tibial tumors of DU 145 (Left) and PC-3 (right) cells. n=3 tumors for each group of PC-3 cells. For DU 145 cells, n=2, 6, 3 and 6 tumors for EV, KLF5, KR and KQ respectively. One representative field per tumor was used for statistical analysis in e-h." Two tailed student t-test was used between two groups, which has been described in the last paragraph of the Method section and at the end of each relevant figure legend.

Comment 3: In figure 3 data shown with the KR and KQ mutants illustrate that subcutaneous tumors expressing the KR mutant grow faster for both DU145 and PC3 cells expressing KQ. Given these differences, one would expect that the transcriptional analysis data would come from tumors growing in the tibia--to ensure that the correct microenvironment is reproduced with analyzing the significance of the KR and Q mutants. However, this does not appear to be the case. Figure 4 states that cells and not tibial tumors were analyzed. If indeed these analyses were performed on cultured cells, the significance of the subsequent studies is questionable. Specifically Figure 4 G-J and Figure 5 A-J.

Response: This is a great question. We totally agree with the reviewer and in fact, we considered using tibial tumors, because this design is definitely more powerful and relevant to mimic the correct microenvironment of bone metastasis. However, isolation of pure tumor cells from mouse tibia is still a huge technical challenge, and it is important to use pure tumor cells for facilitating the accuracy of RNA-Seq analyses. We thus had to choose an alternative strategy: do RNA sequencing using pure tumor cells from culture, screen candidates based on *in vitro* data, and then validate them in tumors from the bone microenvironment. Both PC-3 and DU 145 cells secrete TGF- β , so using *in vitro* cultured cells is still relevant to provide clues. Our successful validation of genes identified from cultured cells for their consistent changes in tibial tumors in fact validate our approaches.

We would like to emphasize the fact that PC-3 and DU 145 cells with the KLF5^{KQ} mutant grew slower than those with the KLF5^{KR} mutant not only in subcutaneous tumors but also in

tibia tumors (New Fig. 2g, h), which suggests that the promoting effects of Ac-KLF5 on bone metastatic lesions is attributed to reasons other than cell proliferation.

To elucidate the mechanisms by which Ac-KLF5 promotes bone metastatic lesions, we employed a co-culture system and a conditioned media (CM) system to test the association between Ac-KLF5 and osteoclast differentiation, a driving factor of bone metastatic lesions. Old Figure 4 D-J (corresponding to New Fig. 4c-h) actually validated the induction of osteoclast differentiation by Ac-KLF5 *in vitro*, which is consistent with the data from the *in vivo* tibia injection model (New Fig. 4a, b). The data in New Figure 5 is the *in vitro* screening for and validation of the potential downstream targets of Ac-KLF5 that mediate Ac-KLF5's function in osteoclast differentiation. Eventually, CXCR4 was identified and further validated as a downstream target of Ac-KLF5 in both *in vitro* and *in vivo* tibia analyses in the New Figure 8. Therefore, New Figures 4 and 5 are necessary for the final *in vivo* confirmation of CXCR4 as a target of Ac-KLF5 in mediating metastatic tumor growth in bone. Use of the co-culture system and conditioned media (CM) system *in vitro* to mimic osteoclast differentiation is also well accepted in the field of bone metastasis research^{35, 36}.

Comment 4: In figure 6 A and B --are tibial tumors or cultured cells analyzed for CXCR4 expression? Is the D analysis from cultured cells or tumor? The relevance of the studies in RAW264.7 cells to tibial osteoclasts is questionable. What are the intratibial levels of IL-11 with or without loss of CXCR4 in the PC3 and DU 145 cells? Also it is hard to believe that cells can sense the difference between 510 and 430 pb/ml of IL-11 based upon classic Scatchard analysis of binding to the receptor. While add back IL-11 experiments were performed, it would be more valid to KO the IL-11 receptor and determine effects.

Response: In old Figure 6A and B (corresponding to New Fig. 6a, c), cultured cells were analyzed for CXCR4 expression, because the upregulation of CXCR4 was found by RNA-Seq and ChIP-Seq by using *in vitro* 2D cultured cells, and these two figures provide the validation in the same cells. Similarly, the data in old Figure 6D (corresponding to New Fig. 6f) were also from cultured cells. We have clearly defined this information in the figure legends. The RAW264.7 cells are of monocyte/macrophage like cell lineage, and extensive studies have proved RAW264.7 cells could respond to stimuli *in vitro* to subsequently generate multinucleated cells with the hallmark characteristics of fully differentiated osteoclasts³⁷. Again use of RAW264.7 cells in co-culture and CM systems is well accepted for investigating the players of osteoclast differentiation^{35, 36}.

For the suggestion to detect the intratibial level of IL-11 with or without CXCR4 in PC-3 and DU 145 cells, we appreciate it and have detected IL-11 by using IHC in tibial samples bearing PCa cells with KR and KQ mutants under the treatment of CXCR4 inhibitor AMD3100 (New Supplementary Fig. 9d, e, shown on *next page*). As we now describe in the Result section (Line 410 on page 15 in the revised manuscript), "We also detected the *in vivo* expression levels of IL-11 in tibial tumors by IHC staining and found that inhibition of CXCR4 by AMD3100 selectively suppressed IL-11 expression levels in KLF5^{KQ} tibial tumors (Supplementary Fig. 9d, e), which further supports the conclusion that IL-11 is a paracrine signaling at the downstream of Ac-KLF5/CXCR4 axis in mediating osteoclast differentiation."

Supplementary figure 9

We also agree with the reviewer that the changes of IL-11 level in DU 145 cells by KLF5 acetylation was subtle (New Fig. 7g). However, addition of 80 pg/ml IL-11 definitely increased the osteoclast differentiation of RAW264.7 cells in the conditioned media (CM) assay from DU 145 KLF5^{KR} cells (New Fig. 7i). The change of IL-11 was more obvious in PC-3 cells (New Fig. 7h). Another point to clarify is that IL-11 is not the only paracrine signaling that mediates the promoting effects of Ac-KLF5 on osteoclast differentiation. For example, SHH also functions in osteoclast differentiation³⁶ and its induction by Ac-KLF5 was also validated in this study (New Fig. 7b, 7c, 9e and Supplementary Fig. 8g, h).

More convincingly, we have knocked down IL-11 to test its role in Ac-KLF5/CXCR4 mediated osteoclast differentiation (New Supplementary Fig. 8e, f, shown *below*). As we now describe in the Result section (Line 349 on Page 14 in the revised manuscript), “knockdown of IL-11 not only abolished the induction of osteoclast differentiation by KLF5^{KQ} (Supplementary Fig. 8e), it also eliminated the suppressive effects of CXCR4 knockdown on osteoclast differentiation (Supplementary Fig. 8f).”

Supplementary figure 8

Comment 5: In Figure 8 A the stated Ac-KLF5 staining is not apparent at the arrows indicated. It is challenging to distinguish the DAB staining from the nuclear hematoxylin. Increased power is needed.

Response: We have included the images with higher magnification in the main figure to better demonstrate the results (New Fig. 9a, shown *below*). The original images with lower magnification have been moved to the New Supplementary Figure 10a, in which the regions enlarged in the New Figure 9a are indicated by black rectangles.

Comment 6: The discussion is way too long.

Response: We have made changes according to this suggestion.

Reference

1. Guo P, *et al.* Pro-proliferative factor KLF5 becomes anti-proliferative in epithelial homeostasis upon signaling-mediated modification. *J Biol Chem* **284**, 6071-6078 (2009).
2. Kaighn ME, Narayan KS, Ohnuki Y, Lechner JF, Jones LW. Establishment and characterization of a human prostatic carcinoma cell line (PC-3). *Invest Urol* **17**, 16-23 (1979).
3. Chung LW, Kao C, Sikes RA, Zhou HE. Human prostate cancer progression models and therapeutic intervention. *Hinyokika Kyo* **43**, 815-820 (1997).
4. Campbell JP, Merkel AR, Masood-Campbell SK, Elefteriou F, Sterling JA. Models of bone metastasis. *J Vis Exp*, e4260 (2012).
5. Mizutani K, *et al.* The chemokine CCL2 increases prostate tumor growth and bone metastasis through macrophage and osteoclast recruitment. *Neoplasia* **11**, 1235-1242 (2009).
6. Valta MP, Tuomela J, Bjartell A, Valve E, Vaananen HK, Harkonen P. FGF-8 is involved in bone metastasis of prostate cancer. *Int J Cancer* **123**, 22-31 (2008).
7. Xu W, *et al.* The systemic delivery of an oncolytic adenovirus expressing decorin inhibits bone metastasis in a mouse model of human prostate cancer. *Gene Ther* **22**, 247-256 (2015).
8. Yang M, *et al.* A fluorescent orthotopic bone metastasis model of human prostate cancer. *Cancer Res* **59**, 781-786 (1999).
9. Fournier PG, *et al.* The TGF-beta Signaling Regulator PMEPA1 Suppresses Prostate Cancer Metastases to Bone. *Cancer Cell* **27**, 809-821 (2015).
10. Conley-LaComb MK, Saliganan A, Kandagatla P, Chen YQ, Cher ML, Chinni SR. PTEN loss mediated Akt activation promotes prostate tumor growth and metastasis via CXCL12/CXCR4 signaling. *Mol Cancer* **12**, 85 (2013).
11. Fisher JL, Schmitt JF, Howard ML, Mackie PS, Choong PF, Risbridger GP. An in vivo model of prostate carcinoma growth and invasion in bone. *Cell Tissue Res* **307**, 337-345 (2002).
12. Hu Z, *et al.* Systemic delivery of oncolytic adenoviruses targeting transforming growth factor-beta inhibits established bone metastasis in a prostate cancer mouse model. *Hum Gene Ther* **23**, 871-882 (2012).
13. Stone KR, Mickey DD, Wunderli H, Mickey GH, Paulson DF. Isolation of a human prostate carcinoma cell line (DU 145). *Int J Cancer* **21**, 274-281 (1978).

14. Ogawa Y, *et al.* Heat shock protein 70 (HSP70) does not prevent the inhibition of cell growth in DU-145 cells treated with TGF-beta1. *Anticancer Res* **21**, 3341-3347 (2001).
15. Ritchie CK, Andrews LR, Thomas KG, Tindall DJ, Fitzpatrick LA. The effects of growth factors associated with osteoblasts on prostate carcinoma proliferation and chemotaxis: implications for the development of metastatic disease. *Endocrinology* **138**, 1145-1150 (1997).
16. Thalmann GN, *et al.* Androgen-independent cancer progression and bone metastasis in the LNCaP model of human prostate cancer. *Cancer Res* **54**, 2577-2581 (1994).
17. Lin DL, *et al.* Bone metastatic LNCaP-derivative C4-2B prostate cancer cell line mineralizes in vitro. *Prostate* **47**, 212-221 (2001).
18. Dai Y, *et al.* The TGF-beta signalling negative regulator PICK1 represses prostate cancer metastasis to bone. *Br J Cancer* **117**, 685-694 (2017).
19. Jin R, *et al.* Activation of NF-kappa B signaling promotes growth of prostate cancer cells in bone. *PLoS ONE* **8**, e60983 (2013).
20. Koreckij TD, *et al.* HE3235 inhibits growth of castration-resistant prostate cancer. *Neoplasia* **11**, 1216-1225 (2009).
21. Lee C, *et al.* Dual targeting c-met and VEGFR2 in osteoblasts suppresses growth and osteolysis of prostate cancer bone metastasis. *Cancer Lett* **414**, 205-213 (2018).
22. Liang W, *et al.* Targeting cathepsin K diminishes prostate cancer establishment and growth in murine bone. *J Cancer Res Clin Oncol* **145**, 1999-2012 (2019).
23. Liao J, Schneider A, Datta NS, McCauley LK. Extracellular calcium as a candidate mediator of prostate cancer skeletal metastasis. *Cancer Res* **66**, 9065-9073 (2006).
24. Lu Y, *et al.* Monocyte chemotactic protein-1 mediates prostate cancer-induced bone resorption. *Cancer Res* **67**, 3646-3653 (2007).
25. Lu Y, *et al.* Activation of MCP-1/CCR2 axis promotes prostate cancer growth in bone. *Clin Exp Metastasis* **26**, 161-169 (2009).
26. Xing C, Fu X, Sun X, Guo P, Li M, Dong JT. Different expression patterns and functions of acetylated and unacetylated Klf5 in the proliferation and differentiation of prostatic epithelial cells. *PLoS One* **8**, e65538 (2013).
27. Zhang B, *et al.* Klf5 acetylation regulates luminal differentiation of basal progenitors in prostate development and regeneration. *Nat Commun* **In press**, (2020).
28. Li X, *et al.* Interruption of KLF5 acetylation converts its function from tumor suppressor to tumor promoter in prostate cancer cells. *Int J Cancer* **136**, 536-546 (2015).

29. Liu R, *et al.* Mifepristone Suppresses Basal Triple-Negative Breast Cancer Stem Cells by Down-regulating KLF5 Expression. *Theranostics* **6**, 533-544 (2016).
30. Maehara O, *et al.* A pivotal role of Kruppel-like factor 5 in regulation of cancer stem-like cells in hepatocellular carcinoma. *Cancer Biol Ther* **16**, 1453-1461 (2015).
31. Dong Z, Yang L, Lai D. KLF5 strengthens drug resistance of ovarian cancer stem-like cells by regulating survivin expression. *Cell proliferation* **46**, 425-435 (2013).
32. David CJ, *et al.* TGF-beta Tumor Suppression through a Lethal EMT. *Cell* **164**, 1015-1030 (2016).
33. Watson PA, Arora VK, Sawyers CL. Emerging mechanisms of resistance to androgen receptor inhibitors in prostate cancer. *Nat Rev Cancer* **15**, 701-711 (2015).
34. Xing C, *et al.* Klf5 deletion promotes Pten deletion-initiated luminal-type mouse prostate tumors through multiple oncogenic signaling pathways. *Neoplasia* **16**, 883-899 (2014).
35. Sethi N, Dai X, Winter CG, Kang Y. Tumor-derived JAGGED1 promotes osteolytic bone metastasis of breast cancer by engaging notch signaling in bone cells. *Cancer Cell* **19**, 192-205 (2011).
36. Wu JB, *et al.* MAOA-Dependent Activation of Shh-IL6-RANKL Signaling Network Promotes Prostate Cancer Metastasis by Engaging Tumor-Stromal Cell Interactions. *Cancer Cell* **31**, 368-382 (2017).
37. Xu XY, *et al.* Differential effects of mechanical strain on osteoclastogenesis and osteoclast-related gene expression in RAW264.7 cells. *Molecular medicine reports* **6**, 409-415 (2012).

REVIEWERS' COMMENTS

Reviewer #1 (Remarks to the Author):

The work for the first time addressed in vivo function of Ac-KLF5 that is induced by TGF-beta, and could contribute to the research field on TGF-beta and cancer progression. In this revised manuscript, the authors well addressed my concerns. I have no additional comments.

Reviewer #2 (Remarks to the Author):

The authors have conscientiously addressed many of my concerns, although the fact has not changed that much of the studies was conducted using long-term cell line models.

Reviewer #3 (Remarks to the Author):

The authors have addressed my concerns. While inclusion of analysis of tumor cells from the tibia would have strengthened the analysis, I understand that this is a challenging experiment. However, single cell-RNAseq analysis would likely provide the level of specificity for tumor cells needed, or FACS sorting of the tumor cells. The authors have chosen an alternate path, while not as strong, is adequate.

Point by point responses to reviewers' comments

Manuscript NCOMMS-19-33932A-Z, by Zhang et al., entitled "Acetylation of KLF5 causes chemoresistant bone metastasis by activating CXCR4/IL-11 while maintaining EMT and tumorigenicity in prostate cancer"

Reviewer #1 (Remarks to the Author):

The work for the first time addressed in vivo function of Ac-KLF5 that is induced by TGF-beta, and could contribute to the research field on TGF-beta and cancer progression. In this revised manuscript, the authors well addressed my concerns. I have no additional comments.

Response: We appreciate the reviewer's approval and positive comments.

Reviewer #2 (Remarks to the Author):

The authors have conscientiously addressed many of my concerns, although the fact has not changed that much of the studies was conducted using long-term cell line models.

Response: We agree with the reviewer that using patient-derived xenograft (PDX) models would provide additional preclinical information and better support for this study. However, we would like to reemphasize that our goal in this study is to understand the role of KLF5 acetylation in the metastatic bone growth of prostate cancer, and the cell line models are relevant to a large extent. For the use of PDX models to mimic prostate cancer bone metastasis, there are also some limitations. For example, single prostate cancer cells have to be isolated from prostate tissues for tibial injection, which would make a tibial PDX model more similar to that of a cell line. In addition, manipulating KLF5 acetylation would be much more challenging than using cell lines unless there is a PDX that lacks KLF5. Avoiding establishment of isogenic cells in PDX models, we could not provide a bona fide control for acetylated KLF5 expression, i.e. KLF5^{K369R} mutant. Alternatively, isogenic cell lines expressing different KLF5 forms could be better than the PDX models for testing the role of KLF5 acetylation in prostate cancer bone metastasis.

Reviewer #3 (Remarks to the Author):

The authors have addressed my concerns. While inclusion of analysis of tumor cells from the tibia would have strengthened the analysis, I understand that this is a challenging experiment. However, single cell-RNAseq analysis would likely provide the level of specificity for tumor cells needed, or FACS sorting of the tumor cells. The authors have chosen an alternate path, while not as strong, is adequate.

Response: We appreciate the reviewer's approval of this manuscript and suggestion of single-cell sequencing.